# Understanding Gradient Orthogonalization for Deep Learning via Non-Euclidean Trust-Region Optimization

## Abstract

Optimization with matrix gradient orthogonalization has recently demonstrated impressive results in the training of deep neural networks (Jordan et al., 2024; Liu et al., 2025). In this paper, we provide a theoretical analysis of this approach. In particular, we show that the orthogonalized gradient method can be seen as a first-order trust-region optimization method, where the trust-region is defined in terms of the matrix spectral norm. Motivated by this observation, we develop the stochastic non-Euclidean trust-region gradient method with momentum, which recovers the Muon optimizer (Jordan et al., 2024) as a special case, along with normalized SGD and signSGD with momentum (Cutkosky & Mehta, 2020; Sun et al., 2023). In addition, we prove state-of-the-art convergence results for the proposed algorithm in a range of scenarios, which involve arbitrary non-Euclidean norms, constrained and composite problems, and non-convex, star-convex, first- and second-order smooth functions. Finally, our theoretical findings provide an explanation for several practical observations, including the practical superiority of Muon compared to the Orthogonal-SGDM algorithm of Tuddenham et al. (2022) and the importance of weight decay in the training of large-scale language models.

## 1 Introduction

Over the past couple of decades, a substantial amount of optimization research has been dedicated to adaptive gradient optimization algorithms (Duchi et al., 2011; Tieleman, 2012; Kingma & Ba, 2014; Gupta et al., 2018; Reddi et al., 2019), with the primary application being the training of deep neural networks (LeCun et al., 2015). One of the most notable results of this line of research is the development of the AdamW optimizer (Loshchilov & Hutter, 2017; Zhuang et al., 2022), which has become the standard algorithm of choice for training large language models (LLMs) (Achiam et al., 2023; Liu et al., 2024; Grattafiori et al., 2024; Anil et al., 2023). However, recently, Jordan et al. (2024); Liu et al. (2025) have made significant progress in the ambitious task of surpassing AdamW in training LLMs using the idea of neural network optimization with orthogonalized gradients (Tuddenham et al., 2022; Jordan et al., 2024). In our paper, we aim to establish theoretical foundations for this promising research direction. Formally speaking, we consider the following composite optimization problem:

$$\min_{x \in \mathcal{X}} \left[ F(x) = f(x) + R(x) \right], \qquad (1)$$

where $\mathcal{X}$ is a finite-dimensional vector space endowed with the inner product $\langle \cdot, \cdot \rangle \colon \mathcal{X} \times \mathcal{X} \to \mathbb{R}$, $f(\cdot) \colon \mathcal{X} \to \mathbb{R}$ is a bounded from below and differentiable objective function, and $R(\cdot) \colon \mathcal{X} \to \mathbb{R} \cup \{+\infty\}$ is a proper,[1] closed, and convex regularizer. Our goal is to study the convergence properties of gradient methods for solving problem (1) in the *stochastic non-Euclidean smooth* setting. We provide a formal description of this setting below and justify our interest in this setting in the upcoming Sections 1.1 and 1.2.

**Stochastic gradient estimator.** We assume access to a stochastic estimator $g(\cdot\,; \xi) \colon \mathcal{X} \to \mathcal{X}$ of the gradient $\nabla f(\cdot)$, where $\xi \sim \mathcal{D}$ is a random variable sampled from a probability distribution $\mathcal{D}$. We

---

[1] Function $R(x)$ is called proper if there exists $x \in \mathcal{X}$ such that $R(x)$ is finite.

assume that the stochastic gradient estimator $g(\cdot; \xi)$ is unbiased and has bounded variance, that is, the following relations hold:

$$\mathbb{E}_{\xi \sim \mathcal{D}}\left[g(x; \xi)\right] = \nabla f(x) \quad \text{and} \quad \mathbb{E}_{\xi \sim \mathcal{D}}\left[\|g(x; \xi) - \nabla f(x)\|_2^2\right] \leq \sigma^2 \quad \text{for all } x \in \mathcal{X}, \quad \text{(A1)}$$

where $\sigma > 0$ is a positive variance parameter, and $\|\cdot\|_2$ is the standard Euclidean norm induced by the inner product $\langle \cdot, \cdot \rangle$, i.e., $\|x\|_2 = \sqrt{\langle x, x \rangle}$. These assumptions have been widely adopted for the analysis of many stochastic gradient optimization algorithms (Ghadimi & Lan, 2013; 2016; Cutkosky & Mehta, 2020; Sun et al., 2023; Horváth et al., 2023; Gorbunov et al., 2020).

**Non-Euclidean norm setting and Lipschitz continuous gradient.** We assume that vector space $\mathcal{X}$ is equipped with a norm $\|\cdot\|: \mathcal{X} \to \mathbb{R}_+$, which possibly does not coincide with the Euclidean norm $\|\cdot\|_2$. In addition, we assume that the gradient $\nabla f(\cdot)$ is Lipschitz continuous with respect to the norm $\|\cdot\|$, that is, the following inequality holds:

$$\|\nabla f(x) - \nabla f(x')\|_* \leq L\|x - x'\| \quad \text{for all } x, x' \in \mathcal{X}, \quad \text{(A2)}$$

where $L > 0$ is the gradient Lipschitz constant, and $\|\cdot\|_*: \mathcal{X} \to \mathbb{R}_+$ is the dual norm associated with $\|\cdot\|$, i.e., $\|x\|_* = \sup_{\|x'\| \leq 1} \langle x, x' \rangle$ for all $x \in \mathcal{X}$. The assumption of gradient Lipschitz continuity is also widespread in the analysis of first-order optimization methods (Ghadimi & Lan, 2013; Gower et al., 2019; Cutkosky & Mehta, 2020; Horváth et al., 2023; Gorbunov et al., 2020). It is important to highlight that while Assumption (A2) uses the dual norm $\|\cdot\|_*$ to measure the difference between the gradients, the variance in Assumption (A1) is measured with respect to the Euclidean norm $\|\cdot\|_2^2$, which is necessary to properly utilize the unbiasedness property of the stochastic gradient estimator $g(\cdot; \xi)$. Therefore, we need to provide a connection between these norms using the following inequality:

$$\|x\|_* \leq \rho \cdot \|x\|_2 \quad \text{for all } x \in \mathcal{X}, \quad \text{(A3)}$$

where $\rho > 0$ is a positive constant. Note that such a constant always exists due to the norm equivalence theorem, which always holds in the finite-dimensional space $\mathcal{X}$.

## 1.1 Main Motivation: Optimization with Orthogonalized Gradients

The main motivation for this work is the recently proposed idea of using *orthogonalized gradient updates* for the training of deep neural networks (Tuddenham et al., 2022; Jordan et al., 2024). To explain the idea, we consider the problem of minimizing a differentiable matrix function $F(\cdot): \mathbb{R}^{m \times n} \to \mathbb{R}$

$$\min_{\mathbf{X} \in \mathbb{R}^{m \times n}} F(\mathbf{X}), \quad (2)$$

which is clearly a special instance of the main problem (1), as long as we choose $\mathcal{X}$ to be the space of $m \times n$ matrices, $\mathcal{X} = \mathbb{R}^{m \times n}$, and assume zero regularization, $R \equiv 0$. The iterations of the simplest version of the orthogonalized gradient method for solving problem (2) can be written as follows:

$$\mathbf{X}_{k+1} = \mathbf{X}_k - \eta \mathbf{O}_k, \quad \mathbf{O}_k = \text{orth}(\mathbf{G}_k), \quad \mathbf{G}_k = \nabla F(\mathbf{X}_k). \quad (3)$$

where $\eta > 0$ is the stepsize, and $\text{orth}(\cdot): \mathbb{R}^{m \times n} \to \mathbb{R}^{m \times n}$ is the orthogonalization function, which is defined as follows:

$$\text{orth}(\mathbf{G}) = (\mathbf{G}\mathbf{G}^\top)^{\frac{\ddagger}{2}}\mathbf{G} \quad \text{for all} \quad \mathbf{G} \in \mathbb{R}^{m \times n}, \quad (4)$$

where $\frac{\ddagger}{2}$ denotes the square root of the pseudoinverse matrix. The name ¡¡orthogonalized¿¿ is attributed to the fact that in the case where the gradient $\nabla F(\mathbf{X}_k)$ is a square and full-rank matrix, the update $\mathbf{O}_k$ can be equivalently expressed as $\mathbf{O}_k = \mathbf{U}_k \mathbf{V}_k^\top$, where $\mathbf{U}_k$ and $\mathbf{V}_k$ are the orthogonal matrix parts of the singular value decomposition of the gradient, i.e., $\nabla F(\mathbf{X}_k) = \mathbf{U}_k \mathbf{\Sigma}_k \mathbf{V}_k^\top$, where $\mathbf{\Sigma}_k \in \mathbb{R}^{m \times n}$ is a diagonal matrix with non-negative entries.

Combining iterations (3) with momentum leads to the Muon optimizer (Jordan et al., 2024) for solving the stochastic version of problem (2). It has been empirically shown that this algorithm can considerably outperform AdamW in the training of both small-scale (Jordan et al., 2024) and large-scale (Liu et al., 2025) language models. However, to the best of our knowledge, *the understanding of the effectiveness of Muon is highly limited*. In addition, it is worth mentioning the Orthogonal-SGDM optimizer (Tuddenham et al., 2022), which differs from Muon only in the order in which the momentum and orthogonalization are applied. Unfortunately, Tuddenham et al. (2022) were not able to outperform a well-tuned standard SGD with momentum using their algorithm, and Jordan et al. (2024) reported that Muon performs substantially better in practice than Orthogonal-SGDM. To the best of our knowledge, *an explanation for this phenomenon is missing from the literature*.

## 1.2 SUMMARY OF CONTRIBUTIONS AND RELATED WORK

Motivated by the above discussion, we set the main goal of this paper to develop the theoretical foundations for optimization with orthogonalized gradients. More specifically, we provide the key contributions listed below.[2]

**Gradient orthogonalization as non-Euclidean trust-region optimization.** Jordan et al. (2024); Bernstein & Newhouse (2024) found an interpretation of the orthogonalized gradient method (3) as the standard gradient descent under the non-Euclidean *matrix spectral norm*. However, this interpretation is highly inaccurate, as further discussed in Section 2.3. In Section 2, we develop a *completely new interpretation* of the gradient orthogonalization as a *trust-region gradient method*, where the trust-region is defined in terms of the non-Euclidean spectral norm. This interpretation is accurate because it allows us to recover the iterations (3) exactly. Moreover, using this interpretation, we obtain several meaningful theoretical results, which we describe further. We also discuss the benefits of using the matrix spectral norm in the training of neural networks in Appendix A.1.

**Stochastic non-Euclidean trust-region gradient method with momentum.** Motivated by our new interpretation of gradient orthogonalization, in Section 3, we develop the *stochastic non-Euclidean trust-region gradient method with momentum* (Algorithm 1) for solving problem (1). This algorithm is designed to work with an arbitrary non-Euclidean norm $\|\cdot\|$, and recovers several existing optimization algorithms: normalized SGD with momentum (Cutkosky & Mehta, 2020), signSGD with momentum (Sun et al., 2023), and Muon. We also prove the $1/\varepsilon^4$ iteration complexity of Algorithm 1 for non-convex functions, which yields *the first convergence result for Muon* and matches the existing optimal results for SGD-type methods for non-convex functions (Ghadimi & Lan, 2013; Arjevani et al., 2023). In addition, in Appendix B, we use our results to provide a theoretical explanation of why Muon outperforms its natural contender, Orthogonal-SGDM (Tuddenham et al., 2022), in practice, and justify the subtle differences in the design of these algorithms.

**Convergence analysis for star-convex functions.** In Section 4.1, we develop the *stochastic non-Euclidean trust-region gradient method with weight decay* (Algorithm 2) for solving problem (1) by utilizing the popular weight decay mechanism (Loshchilov & Hutter, 2017). We obtain the improved $1/\varepsilon^3$ iteration complexity of Algorithm 2 under the star-convexity assumption (Nesterov & Polyak, 2006). In addition, in Appendix C, we prove the same $1/\varepsilon^3$ complexity result for Algorithm 1 without weight decay, by additionally assuming the boundedness of $\operatorname{dom} R$. Our results support the practical evidence for the importance of incorporating weight decay into Muon for the training of large-scale language models, and explain why it may be less important for the training of small-scale language models, as reported by Liu et al. (2025); Jordan et al. (2024). Another important remark is that our theoretical results for star-convex functions hold substantial practical interest because there is empirical evidence (Zhou et al., 2019; Kleinberg et al., 2018) suggesting that deep neural networks may adhere to star-convexity or its variants.

**Convergence analysis for second-order smooth functions.** Inspired by the theoretical results of Cutkosky & Mehta (2020), in Section 5, we develop the *stochastic non-Euclidean trust-region gradient method with extrapolation* for solving problem (1) by incorporating a certain extrapolation step into Algorithm 2. We obtain the $1/\varepsilon^{3.5}$ complexity for non-convex functions under the additional second-order smoothness assumption, which matches the results of Cutkosky & Mehta (2020); Sun et al. (2023) for normalized SGD and signSGD and improves upon the standard $1/\varepsilon^4$ complexity for SGD-type methods for non-convex functions (Ghadimi & Lan, 2013; Arjevani et al., 2023). In addition, we provide a convergence analysis for Algorithm 3 for star-convex functions, which also shows an improvement over the results for Algorithms 1 and 2.

## 2 NON-EUCLIDEAN TRUST-REGION GRADIENT METHOD

### 2.1 GRADIENT DESCENT AS MAJORIZATION-MINIMIZATION

It is generally known that the standard proximal gradient descent can be seen as an instance of the majorization-minimization (MM) algorithm (Hunter & Lange, 2004). In particular, we can replace

---

[2]In Appendix A.2 we compare our results with the concurrent work of Pethick et al. (2025), which we found only after the initial version of our paper appeared online.

the objective function $F(x)$ in problem (1) with the following approximation at an arbitrary point $z \in \mathcal{X}$:

$$\mathcal{U}(x; z) = f(z) + \langle \nabla f(z), x - z \rangle + \frac{1}{2\theta} \|x - z\|^2 + R(x), \tag{5}$$

which is constructed by replacing function $f(x)$ with its first-order Taylor approximation at point $z$ and adding the squared norm regularization $\frac{1}{2\theta} \|x - z\|^2$. This approximation is accurate at point $z$, i.e., $F(z) = \mathcal{U}(z; z)$, and using standard arguments, one can show that $\mathcal{U}(x; z)$ majorizes $F(x)$, i.e., $F(x) \leq \mathcal{U}(x; z)$ for all $x \in \mathcal{X}$, as long as $\theta \geq 1/L$ and Assumption (A2) holds. Consequently, the iterations of the MM algorithm can be written as follows:

$$x_{k+1} = \operatorname*{argmin}_{x \in \mathcal{X}} \mathcal{U}(x; x_k). \tag{6}$$

One can immediately observe that these iterations coincide with the proximal gradient method (Nesterov, 2013) as long as we assume the Euclidean norm setting, i.e., $\|\cdot\| = \|\cdot\|_* = \|\cdot\|_2$:

$$x_{k+1} = \operatorname{prox}_{\theta R(\cdot)}(x_k - \theta \nabla f(x_k)). \tag{7}$$

The convergence guarantees for this algorithm were established in the Euclidean setting for both non-convex and convex functions $f(x)$ by Nesterov (2013).

## 2.2 Non-Euclidean Trust-Region Gradient Method

The trust-region optimization approach (Conn et al., 2000; Jiang et al., 2023) is a notable alternative to the MM principle discussed above. It is usually employed when access to the second-order derivatives of the objective function is assumed. This approach involves minimizing a second-order Taylor-like approximation of the objective function at a given point within a certain neighborhood of that point, which is called the trust-region.

Further, we construct a first-order, i.e., gradient, trust-region optimization algorithm for solving problem (1). More specifically, we replace the objective function $F(x)$ with the following approximation at an arbitrary point $z \in \mathcal{X}$:

$$\mathcal{A}(x; z) = f(z) + \langle \nabla f(z), x - z \rangle + R(x), \tag{8}$$

which does not necessarily majorize function $F(x)$ due to the lack of squared norm regularization, in contrast to $\mathcal{U}(x; z)$ in eq. (5). Thus, the iterations of the trust-region method minimize this approximation only within the ball of radius $\eta > 0$ defined in terms of the non-Euclidean norm $\|\cdot\|$ as follows:

$$x_{k+1} = \operatorname*{argmin}_{x \in \mathcal{X}} \mathcal{A}(x; x_k) \quad \text{s.t.} \quad \|x - x_k\| \leq \eta. \tag{9}$$

As a particular example, it is easy to verify that in the Euclidean norm setting with zero regularization, i.e., where $\|\cdot\| = \|\cdot\|_* = \|\cdot\|_2$ and $R \equiv 0$, these iterations coincide with normalized gradient descent (Shor, 2012):

$$x_{k+1} = x_k - \frac{\eta}{\|\nabla f(x_k)\|_2} \nabla f(x_k). \tag{10}$$

The convergence guarantees for the iterations (10) were established in the convex case by Grimmer (2019) and in some non-convex scenarios by Nesterov (1984). However, to the best of our knowledge, convergence results for the trust-region gradient method (9) with a general non-Euclidean norm $\|\cdot\|$ and an arbitrary proper closed and convex regularizer $R(\cdot)$ are unknown for both convex and non-convex functions $f(x)$. We provide such results for non-convex functions in Section 2.4 and for star-convex functions in Appendix C.

## 2.3 Gradient Orthogonalization for Matrix Function Optimization

Now, we revisit the minimization problem (2) over the space of matrices $\mathcal{X} = \mathbb{R}^{m \times n}$, which is a special instance of the main problem (1) with zero regularization $R \equiv 0$. We choose the non-Euclidean norms $\|\cdot\|$ and $\|\cdot\|_*$ to be the spectral $\|\cdot\|_{\mathrm{op}}$ and the nuclear $\|\cdot\|_{\mathrm{nuc}}$ matrix norms, respectively. That is, for an arbitrary matrix $\mathbf{X} \in \mathbb{R}^{m \times n}$, we have $\|\mathbf{X}\|_{\mathrm{op}} = \|\boldsymbol{\sigma}(\mathbf{X})\|_\infty$ and $\|\mathbf{X}\|_{\mathrm{nuc}} = \|\boldsymbol{\sigma}(\mathbf{X})\|_1$, where $\boldsymbol{\sigma}(\mathbf{X})$ is the vector of singular values of $\mathbf{X}$. Thus, one can verify that *the non-Euclidean trust-region gradient method* (9) *is exactly equivalent to the orthogonalized gradient method* (3). This is one of the key observations of our work. It is in sharp contrast with Bernstein & Newhouse

(2024), who suggested the interpretation of the orthogonalized gradient method as the gradient descent (6) in the same non-Euclidean setting. As mentioned in Section 1.2, this interpretation is highly inaccurate because the iterations (6) reduce to the following:

$$\mathbf{X}_{k+1} = \mathbf{X}_k - \theta \|\mathbf{G}_k\|_{\mathrm{nuc}} \mathbf{O}_k, \tag{11}$$

where $\mathbf{G}_k = \nabla F(\mathbf{X}_k)$, and $\mathbf{O}_k \in \mathbb{R}^{m \times n}$ is defined in eq. (3). Indeed, there is an extra nuclear norm factor $\|\mathbf{G}_k\|_{\mathrm{nuc}}$ in the update (11). More importantly, in contrast to this interpretation of Bernstein & Newhouse (2024), we can use our trust-region approach to establish convergence guarantees for Muon.

## 2.4 Convergence Analysis for Non-Convex Functions

In this section, we provide a convergence analysis for the non-Euclidean trust-region gradient method (9) for solving the non-stochastic version of problem (1) with the non-convex objective function $f(x)$ in the following Theorem 1 and Corollary 1. The proof is available in Appendix D.1.

**Theorem 1.** *Let Assumption (A2) hold and let $x_0 \in \mathrm{dom}\, R$. Then the iterations (9) satisfy the following inequality:*

$$\min_{k=1,\dots,K} \|\nabla f(x_k) + \hat{\nabla} R_k\|_* \le \frac{\Delta_0}{\eta K} + \frac{3L\eta}{2}, \tag{12}$$

*where $\hat{\nabla} R_k \in \partial R(x_k)$, $\Delta_0 = F(x_0) - \inf_x F(x)$.*

**Corollary 1.** *To reach the precision $\min_{k=1\dots K} \|\nabla f(x_k) + \hat{\nabla} R_k\|_* \le \varepsilon$ by the iterations (9) under the conditions of Theorem 1, it is sufficient to choose the stepsize $\eta$ and the number of iterations $K$ as follows:*

$$\eta = \mathcal{O}\left(\frac{\varepsilon}{L}\right), \qquad K = \mathcal{O}\left(\frac{L\Delta_0}{\varepsilon^2}\right). \tag{13}$$

We make two remarks regarding Theorem 1 and Corollary 1. First, the generalized stationarity condition $\|\nabla f(x_k) + \hat{\nabla} R_k\|_* \le \varepsilon$ is widely adopted in the literature (Nesterov, 2013). In the case of zero regularization, $R \equiv 0$, it reduces to the standard definition of an $\varepsilon$-stationary point, $\|\nabla f(x_k)\|_* \le \varepsilon$. Second, the obtained iteration complexity $K = \mathcal{O}\left(L\Delta_0/\varepsilon^2\right)$ matches the standard result for gradient descent in the Euclidean setting by Nesterov (2013) and cannot be improved in general (Carmon et al., 2020).

## 3 Stochastic Non-Euclidean Trust-Region Gradient Method with Momentum

### 3.1 The Algorithm

In this section, we present Algorithm 1 for solving problem (1) in the stochastic setting. We call this algorithm the stochastic non-Euclidean trust-region gradient method with momentum. The main idea is to replace the gradient $\nabla f(x_k)$ in the trust-region gradient method (9) with the momentum term $m_{k+1}$, which is updated according to eq. (14). While the idea is inspired by the results for normalized SGD with momentum by Cutkosky & Mehta (2020), their analysis is not applicable in the case of a non-Euclidean norm $\|\cdot\|$ and a nonzero regularizer $R \not\equiv 0$. We provide the convergence analysis of Algorithm 1 for non-convex functions in Section 3.2 and for star-convex functions in Appendix C.

It is also important to highlight that by choosing different norms $\|\cdot\|$, Algorithm 1 can be reduced to several important special instances: **(i)** Euclidean norm, $\|\cdot\| = \|\cdot\|_2$ – normalized SGD with momentum (Cutkosky & Mehta, 2020); **(ii)** infinity-norm, $\|\cdot\| = \|\cdot\|_\infty$ – signSGD with momentum (Sun et al., 2023); **(iii)** spectral norm, $\|\cdot\| = \|\cdot\|_{\mathrm{op}}$ – Muon (Jordan et al., 2024). The latter reduction is implied by the discussion in Section 2.3.

### 3.2 Convergence Analysis for Non-Convex Functions

In this section, we provide the convergence analysis for Algorithm 1 for solving problem (1) in the stochastic non-convex case in the following Theorem 2. The proof is available in Appendix D.2.

---

**Algorithm 1** Stochastic Non-Euclidean Trust-Region Gradient Method with Momentum

1: **input:** $x_0, m_0 \in \mathcal{X}$
2: **parameters:** stepsize $\eta > 0$, momentum $\alpha \in (0, 1)$, number of iterations $K \in \{1, 2, \dots\}$
3: **for** $k = 0, 1, \dots, K - 1$ **do**
4:     Sample $\xi_k \sim \mathcal{D}$
5:     Compute $m_{k+1}$ as follows:

$$m_{k+1} = (1 - \alpha)m_k + \alpha g(x_k; \xi_k) \tag{14}$$

6:     Compute $x_{k+1}$ as follows:

$$x_{k+1} = \underset{x \in \mathcal{X}}{\operatorname{argmin}}[\langle m_{k+1}, x \rangle + R(x)] \quad \text{s.t.} \quad \|x - x_k\| \le \eta \tag{15}$$

7: **output:** $x_K \in \mathcal{X}$

---

**Theorem 2.** *Let Assumptions (A1) to (A3) hold, and let $x_0 \in \operatorname{dom} R$ and $m_0 = g(x_0, \xi_0)$. Then the iterations of Algorithm 1 satisfy the following inequality:*

$$\mathbb{E}\left[\min_{k=1,\dots,K} \|\nabla f(x_k) + \hat{\nabla} R_k\|_*\right] \le \frac{\Delta_0}{\eta K} + \frac{2\rho\sigma}{\alpha K} + 2\sqrt{\alpha}\rho\sigma + \frac{7L\eta}{2} + \frac{2L\eta}{\alpha}, \tag{16}$$

*where $\hat{\nabla} R_k \in \partial R(x_k)$, $\Delta_0 = F(x_0) - \inf_x F(x)$.*

**Corollary 2.** *To reach the precision $\mathbb{E}\left[\min_{k=1\dots K} \|\nabla f(x_k) + \hat{\nabla} R_k\|_*\right] \le \varepsilon$ by Algorithm 1 under the conditions of Theorem 2, it is sufficient to choose the parameters of Algorithm 1 as follows:*

$$\eta = \mathcal{O}\left(\min\left\{\frac{\varepsilon}{L}, \frac{\varepsilon^3}{\rho^2\sigma^2 L}\right\}\right), \qquad \alpha = \mathcal{O}\left(\min\left\{1, \frac{\varepsilon^2}{\rho^2\sigma^2}\right\}\right), \tag{17}$$

$$K = \mathcal{O}\left(\max\left\{\frac{\rho\sigma}{\varepsilon}, \frac{\rho^3\sigma^3}{\varepsilon^3}, \frac{L\Delta_0}{\varepsilon^2}, \frac{L\Delta_0\rho^2\sigma^2}{\varepsilon^4}\right\}\right). \tag{18}$$

Using Theorem 2, we obtain the explicit complexity result for Algorithm 1 in Corollary 2. Similarly to the convergence result in Corollary 1 for the deterministic non-Euclidean trust-region gradient method (9), Corollary 2 establishes the convergence result for Algorithm 1 in terms of the generalized expected stationarity. The iteration complexity (18) is proportional to $1/\varepsilon^4$. It matches the existing state-of-the-art results for SGD-type methods (Ghadimi & Lan, 2013; Cutkosky & Mehta, 2020; Sun et al., 2023) and cannot be improved (Arjevani et al., 2023) under Assumptions (A1) to (A3).

Another important remark is that Algorithm 1 with the spectral norm $\|\cdot\| = \|\cdot\|_{\mathrm{op}}$ and zero regularizer $R \equiv 0$ exactly matches Muon. As previously discussed, there is another variant of the orthogonalized gradient method with momentum, namely, Orthogonal-SGDM. While Muon demonstrated strong results in the training of small-scale language models (Jordan et al., 2024), it was reported by Tuddenham et al. (2022); Jordan et al. (2024) that Orthogonal-SGDM performs worse than Muon or a well-tuned SGD with momentum. Our theoretical results may provide a possible explanation for this practical difference. We discuss this in Appendix B.

## 4 ALGORITHMS WITH WEIGHT DECAY FOR STAR-CONVEX FUNCTIONS

### 4.1 NON-EUCLIDEAN TRUST-REGION GRADIENT METHODS WITH WEIGHT DECAY

In this section, we develop non-Euclidean trust-region gradient methods for solving problem (1) in the case where the objective function $f(x)$ is star-convex, that is, the following inequality holds:

$$f(\beta x^* + (1 - \beta)x) \le \beta f(x^*) + (1 - \beta)f(x) \quad \text{for all } x \in \mathcal{X}, \tag{A4}$$

where $x^* \in \mathcal{X}$ is a solution to problem (1). We start with the non-stochastic version of the problem. It turns out that under assumption (A4), it is possible to obtain improved convergence guarantees for the iterations (9) as long as these iterations are bounded. Unfortunately, proving the boundedness of these iterations requires additional assumptions, such as a bounded $\operatorname{dom} R$, or bounded sublevel

---

**Algorithm 2** Stochastic Non-Euclidean Trust-Region Gradient Method with Weight Decay

---

1: **input:** $x_0, m_0 \in \mathcal{X}$
2: **parameters:** $\eta > 0$, $\alpha \in (0, 1)$, $K \in \{1, 2, \ldots\}$, weight decay $\beta \in (0, 1)$
3: **for** $k = 0, 1, \ldots, K - 1$ **do**
4:      Sample $\xi_k \sim \mathcal{D}$
5:      Compute $m_{k+1}$ as follows:

$$m_{k+1} = (1 - \alpha)m_k + \alpha g(x_k; \xi_k) \tag{20}$$

6:      Compute $x_{k+1}$ as follows:

$$x_{k+1} = \operatorname*{argmin}_{x \in \mathcal{X}} [\langle m_{k+1}, x \rangle + R(x)] \quad \text{s.t.} \quad \|x - (1 - \beta)x_k\| \leq \eta \tag{21}$$

7: **output:** $x_K \in \mathcal{X}$

---

sets of the objective function $F(x)$, which often do not hold. We tackle this issue by shifting the center of the trust-region in eq. (9) by a factor of $(1 - \beta)$ towards the zero. This idea leads to the non-Euclidean trust-region gradient method with weight decay (19).

$$x_{k+1} = \operatorname*{argmin}_{x \in \mathcal{X}} \mathcal{A}(x; x_k) \quad \text{s.t.} \quad \|x - (1 - \beta)x_k\| \leq \eta. \tag{19}$$

It is not hard to verify that the iterations (19) are bounded. On the other hand, the original and shifted trust-regions do not differ much. In particular, it is possible to show that they have a non-zero intersection. Hence, the shift modification will not hurt the convergence properties of the iterations (19), which we prove in Section 4.2.

Furthermore, we apply a similar modification to Algorithm 1. This modification leads to Algorithm 2, which we call the stochastic non-Euclidean trust-region gradient method with weight decay. Similar to the discussion in Section 3.1, one can show that by choosing different norms $\|\cdot\|$, Algorithm 2 reduces to the normalized SGD, signSGD, and Muon with momentum and weight decay.

## 4.2 CONVERGENCE ANALYSIS FOR STAR-CONVEX FUNCTIONS

In this section, we provide the convergence analysis for the iterations (19) and Algorithm 2 for solving problem (1) in the star-convex case. We start with the result in Theorem 3 for the deterministic iterations (19). The proof is available in Appendix D.3. Next, we obtain the convergence result for the stochastic Algorithm 2 in Theorem 4. The proof is available in Appendix D.4.

**Theorem 3.** *Let Assumptions (A2) and (A4) hold, and let $x_0 \in \operatorname{dom} R$. Let the parameters $\eta$ and $\beta$ satisfy the following inequality:*

$$\eta \geq \beta \max \{\|x_0\|, \|x^*\|\}. \tag{22}$$

*Then, the iterations* (19) *satisfy the following inequality:*

$$F(x_K) - F(x^*) \leq (1 - \beta)^K (F(x_0) - F(x^*)) + \frac{4L\eta^2}{\beta}. \tag{23}$$

**Theorem 4.** *Let Assumptions (A1) to (A4) hold, and let $x_0 \in \operatorname{dom} R$ and $m_0 = g(x_0, \xi_0)$. Let the parameters $\eta$ and $\beta$ satisfy eq. (22). Then, the output of Algorithm 2 satisfies the following inequality:*

$$\mathbb{E}\left[F(x_K) - F(x^*)\right] \leq (1 - \beta)^K (F(x_0) - F(x^*)) + 2\eta\rho\sigma \left(\frac{1}{\alpha} + \frac{\sqrt{\alpha}}{\beta}\right) + \frac{4L\eta^2}{\beta}\left(1 + \frac{1}{\alpha}\right). \tag{24}$$

Finally, using Theorems 3 and 4, it is not hard to obtain the explicit iteration complexities for the iterations (19) and Algorithm 2 in Corollaries 3 and 4, respectively. The proofs are omitted due to their simplicity.

**Corollary 3.** *Under the conditions of Theorem 3, let $\|x_0\| \leq \|x^*\| = D$. To reach the precision $F(x_K) - F(x^*) \leq \varepsilon$ by the iterations* (19)*, it is sufficient to choose the parameters $\eta$ and $\beta$ and the number of iterations $K$ as follows:*

$$\eta = \beta D, \qquad \beta = \mathcal{O}\left(\min\left\{1, \frac{\varepsilon}{LD}\right\}\right), \qquad K = \tilde{\mathcal{O}}\left(\max\left\{1, \frac{LD^2}{\varepsilon}\right\}\right). \tag{25}$$

**Corollary 4.** *Under the conditions of Theorem 4, let $\|x_0\| \leq \|x^*\| = D$. To reach the precision $\mathbb{E}\left[F(x_K) - F(x^*)\right] \leq \varepsilon$ by Algorithm 2, it is sufficient to choose the parameters as follows:*

$$\alpha = \mathcal{O}\left(\min\left\{1, \frac{\varepsilon^2}{D^2\rho^2\sigma^2}\right\}\right), \quad \beta = \mathcal{O}\left(\min\left\{1, \frac{\varepsilon}{LD^2}, \frac{\varepsilon}{D\rho\sigma}, \frac{\varepsilon^3}{D^3\rho^3\sigma^3}, \frac{\varepsilon^3}{LD^3\rho^2\sigma^2}\right\}\right), \quad (26)$$

$$\eta = \beta D, \qquad\qquad K = \tilde{\mathcal{O}}\left(\max\left\{1, \frac{LD^2}{\varepsilon}, \frac{D\rho\sigma}{\varepsilon}, \frac{D^3\rho^3\sigma^3}{\varepsilon^3}, \frac{LD^3\rho^2\sigma^2}{\varepsilon^3}\right\}\right). \quad (27)$$

Both Corollaries 3 and 4 recover the standard iteration complexity $\mathcal{O}(LD^2/\varepsilon)$ of gradient descent (Nesterov et al., 2018) for solving deterministic convex minimization problems, up to logarithmic factors. This complexity cannot be improved in the general non-Euclidean norm setting (Guzmán & Nemirovski, 2015). In the stochastic setting, Corollary 4 implies complexity proportional to $1/\varepsilon^3$. This improves upon the $1/\varepsilon^{3.5}$ result of Cutkosky & Mehta (2020); Sun et al. (2023), which was obtained only for normalized SGD and signSGD with momentum, and for non-convex second-order smooth functions.[3]

Another important remark is that Algorithm 2 with the spectral norm $\|\cdot\| = \|\cdot\|_{\text{op}}$ and zero regularizer $R \equiv 0$ exactly matches Muon with weight decay, as previously discussed in Section 4.1. It was empirically shown by Liu et al. (2025) that incorporating weight decay in Muon is crucial for outperforming AdamW in the training of large-scale language models. In particular, they reported that without weight decay, the iterations of Muon grow too large and the performance gains of Muon over AdamW diminish. Our theory suggests that combining Muon with weight decay may be the right solution for both issues. Indeed, we prove that the iterations of Algorithm 2 are bounded and obtain the improved complexity $1/\varepsilon^3$ of Algorithm 2 in Corollary 4.

# 5 ALGORITHMS WITH EXTRAPOLATION FOR SECOND-ORDER SMOOTH FUNCTIONS

## 5.1 STOCHASTIC NON-EUCLIDEAN TRUST-REGION GRADIENT METHOD WITH EXTRAPOLATION

In this section, we modify Algorithms 1 and 2 to solve problem (1) in the stochastic setting in the case where the objective function $f(x)$ has an $H$-Lipschitz Hessian with respect to the non-Euclidean norm $\|\cdot\|$, which implies the following inequality:

$$\|(\nabla^2 f(x) - \nabla^2 f(x'))(x - x')\|_* \leq H\|x - x'\|^2 \quad \text{for all } x, x' \in \mathcal{X}, \quad (A5)$$

where $H > 0$ is the Hessian Lipschitz constant. The idea for the modification is inspired by Cutkosky & Mehta (2020). In particular, we replace the stochastic gradient $g(x_k, \xi_k)$ in the update of the momentum term $m_{k+1}$ in eqs. (14) and (20) with the stochastic gradient $g(\overline{x}_k; \xi_k)$ computed at a different point $\overline{x}_k$, which is updated using the extrapolation step (30). This modification leads to Algorithm 3, which we call the stochastic non-Euclidean trust-region gradient method with extrapolation. While we borrow the idea for the analysis of the extrapolation step (30) from Cutkosky & Mehta (2020); Sun et al. (2023), their analysis is limited to normalized SGD and signSGD for non-convex functions with zero regularization $R \equiv 0$. In contrast, in Section 5.2, we provide a convergence analysis for Algorithm 3 with an arbitrary non-Euclidean norm $\|\cdot\|$ and regularizer $R(\cdot)$ in the non-convex and star-convex cases.

Note that, similarly to Algorithm 2 described in Section 4.1, Algorithm 3 makes use of weight decay in eq. (29) along with the extrapolation step (30) when dealing with star-convex functions. Another important remark is that Algorithm 3 with the matrix spectral norm $\|\cdot\|_{\text{op}}$ turns into a new variant of Muon with extrapolation. We discuss this further in Section 5.2.

## 5.2 CONVERGENCE ANALYSIS

We provide the convergence analysis for Algorithm 3 for solving problem (1) with the non-convex objective function in Theorem 5 and with the star-convex objective function in Theorem 6. The proofs of Theorems 5 and 6 are available in Appendices D.5 and D.6, respectively.

---

[3]Additional details on the correctness of this comparison are provided in Appendix A.3.

---

**Algorithm 3** Stochastic Non-Euclidean Trust-Region Gradient Method with Extrapolation

---

1: **input:** $x_0 = \overline{x}_0, m_0 \in \mathcal{X}$
2: **parameters:** $\eta > 0$, $\alpha \in (0,1)$, $K \in \{1, 2, \ldots\}$, $\beta \in (0,1)$, extrapolation $\gamma > 0$
3: **for** $k = 0, 1, \ldots, K-1$ **do**
4:      Sample $\xi_k \sim \mathcal{D}$
5:      Compute $m_{k+1}$ as follows:

$$m_{k+1} = (1-\alpha)m_k + \alpha g(\overline{x}_k; \xi_k) \tag{28}$$

6:      Compute $x_{k+1}$ as follows:

$$x_{k+1} = \operatorname*{argmin}_{x \in \mathcal{X}}[\langle m_{k+1}, x \rangle + R(x)] \quad \text{s.t.} \quad \|x - (1-\beta)x_k\| \le \eta \tag{29}$$

7:      Compute $\overline{x}_{k+1}$ as follows:

$$\overline{x}_{k+1} = x_k + \gamma(x_{k+1} - x_k) \tag{30}$$

8: **output:** $x_K \in \mathcal{X}$

---

**Theorem 5.** *Let Assumptions (A1) to (A3) and (A5) hold, let $x_0 \in \operatorname{dom} R$ and $m_0 = g(x_0, \xi_0)$, and let $\beta = 0$ and $\gamma = 1/\alpha$. Then the iterations of Algorithm 3 satisfy the following inequality:*

$$\mathbb{E}\left[\min_{k=1,\ldots,K}\|\nabla f(x_k) + \hat{\nabla} R_k\|_*\right] \le \frac{\Delta_0}{\eta K} + \frac{7L\eta}{2} + \frac{H\eta^2}{\alpha^2} + \frac{2\rho\sigma}{\alpha K} + 2\sqrt{\alpha}\rho\sigma, \tag{31}$$

*where $\hat{\nabla} R_k \in \partial R(x_k)$, $\Delta_0 = F(x_0) - \inf_x F(x)$.*

**Theorem 6.** *Let Assumptions (A1) to (A5) hold, and let $x_0 \in \operatorname{dom} R$ and $m_0 = g(x_0, \xi_0)$. Let the parameters $\eta$ and $\beta$ satisfy eq. (22), and let $\gamma = 1/\alpha$. Then the output of Algorithm 3 satisfies*

$$\mathbb{E}\left[F(x_K) - F(x^*)\right] \le (1-\beta)^K(F(x_0) - F(x^*)) + 2\eta\rho\sigma\left(\frac{1}{\alpha} + \frac{\sqrt{\alpha}}{\beta}\right) + \frac{4L\eta^2}{\beta} + \frac{4H\eta^3}{\alpha^2\beta}. \tag{32}$$

Using Theorems 5 and 6, it is not hard to obtain the explicit iteration complexities Algorithm 3 in Corollaries 3 and 4 for non-convex and star-convex functions, respectively. The proofs are omitted due to their simplicity.

**Corollary 5.** *To reach the precision $\mathbb{E}\left[\min_{k=1\ldots K}\|\nabla f(x_k) + \hat{\nabla} R_k\|_*\right] \le \varepsilon$ by Algorithm 3 under the conditions of Theorem 5, it is sufficient to choose the parameters as follows:*

$$\eta = \mathcal{O}\left(\min\left\{\frac{\varepsilon}{L}, \frac{\varepsilon^{1/2}}{H^{1/2}}, \frac{\varepsilon^{5/2}}{\rho^2\sigma^2 H^{1/2}}\right\}\right), \quad \alpha = \mathcal{O}\left(\min\left\{1, \frac{\varepsilon^2}{\rho^2\sigma^2}\right\}\right), \quad \beta = 0, \tag{33}$$

$$K = \tilde{\mathcal{O}}\left(\max\left\{\frac{\rho\sigma}{\varepsilon}, \frac{\rho^3\sigma^3}{\varepsilon^3}, \frac{L\Delta_0}{\varepsilon^2}, \frac{H^{1/2}\Delta_0}{\varepsilon^{3/2}}, \frac{H^{1/2}\Delta_0\rho^2\sigma^2}{\varepsilon^{7/2}}\right\}\right). \tag{34}$$

**Corollary 6.** *Under the conditions of Theorem 6, let $\|x_0\| \le \|x^*\| = D$. To reach the precision $\mathbb{E}\left[F(x_K) - F(x^*)\right] \le \varepsilon$ by Algorithm 3, it is sufficient to choose the parameters as follows:*

$$\beta = \mathcal{O}\left(\min\left\{1, \frac{\varepsilon}{LD^2}, \frac{\alpha\varepsilon}{D\rho\sigma}, \frac{\alpha\varepsilon^{1/2}}{H^{1/2}D^{3/2}}\right\}\right), \qquad \alpha = \mathcal{O}\left(\min\left\{1, \frac{\varepsilon^2}{D^2\rho^2\sigma^2}\right\}\right), \tag{35}$$

$$K = \tilde{\mathcal{O}}\left(\max\left\{1, \frac{D\rho\sigma}{\varepsilon}, \frac{D^3\rho^3\sigma^3}{\varepsilon^3}, \frac{LD^2}{\varepsilon}, \frac{H^{1/2}D^{3/2}}{\varepsilon^{1/2}}, \frac{H^{1/2}D^{7/2}\rho^2\sigma^2}{\varepsilon^{5/2}}\right\}\right), \quad \eta = \beta D. \tag{36}$$

Corollary 5 establishes an improved $1/\varepsilon^{3.5}$ iteration complexity compared to the $1/\varepsilon^4$ complexity of Algorithm 1 in Corollary 2, matching the results of Cutkosky & Mehta (2020); Sun et al. (2023) for normalized SGD and signSGD for non-convex second-order smooth functions.

Unfortunately, Algorithm 3 cannot achieve a major improvement over Algorithm 2 in the case of star-convex functions, since both have the term $(\rho\sigma D/\varepsilon)^3$ in their complexities in Corollaries 4 and 6, respectively. However, the practical success of Muon and the discussion in appendix A.1 may indicate that the "geometric" terms, which depend on the Lipschitz constants $L$ and $H$, play a more important role compared to the purely "stochastic" terms, which depend only on the variance $\sigma$. Hence, the improvement in the "mixed" term $\sqrt{\rho^4\sigma^4 HD^7/\varepsilon^5}$ in Corollary 6 over the corresponding term $\rho^2\sigma^2 LD^3/\varepsilon^3$ in Corollary 4 may be significant. Overall, it is an important open theoretical and practical question whether the extrapolation step (30) can improve the convergence of Muon.

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

# A    ADDITIONAL DISCUSSIONS

## A.1    MOTIVITION FOR THE NON-EUCLIDEAN SMOOTHNESS ASSUMPTIONS

In this section, we provide a short discussion on the benefits of using the matrix spectral norm, $\|\cdot\| = \|\cdot\|_{\mathrm{op}}$, in the training of neural networks. In particular, we consider the following simple example of problem (2), which motivates the smoothness Assumptions (A2) and (A5):

$$\min_{\mathbf{X}\in\mathbb{R}^{m\times n}} \left[ F(\mathbf{X}) = \frac{1}{N} \sum_{i=1}^{N} \mathcal{L}_i(\mathbf{X}a_i) \right], \tag{37}$$

where $a_i \in \mathbb{R}^n$ are some vectors, and $\mathcal{L}_i(\cdot)\colon \mathbb{R}^m \to \mathbb{R}$ are loss functions. Our example can be seen as a generalization of the example in Proposition 6 by Bernstein & Newhouse (2024) and is also inspired by some earlier works of Carlson et al. (2015b;a). This example holds interest in the context of deep learning due to the matrix-vector multiplication term $\mathbf{X}a_i$, because similar structures appear in hidden layers of neural networks.

We show that Assumptions (A2) and (A5) are implied by the standard first- and second-order smoothness assumptions of functions $\mathcal{L}_i(\cdot)$ with respect to the standard Euclidean norm $\|\cdot\|_2$ in the following Lemmas 1 and 2, respectively. The proofs are available in Appendices A.4 and A.5. The main observation here is that the smoothness constants $L$ and $H$ depend on the Euclidean norm $\|\mathbf{X}a_i\|_2$, which can be naturally upper-bounded using the spectral norm as $\|\mathbf{X}a_i\|_2 \leq \|\mathbf{X}\|_{\mathrm{op}}\|a_i\|_2$. Note that a similar upper bound can be obtained using the standard Euclidean, aka Frobenius, norm of matrix $\mathbf{X}$, which, however, would be much less tight. Also, note that besides the matrix spectral norm $\|\cdot\|_{\mathrm{op}}$, it is also possible to obtain similar upper bounds using more general operator norms. A more detailed study of this question is provided by Pethick et al. (2025).

**Lemma 1.** *Let each function $\mathcal{L}_i(\cdot)$ in problem (37) be continuously differentiable and have a $\lambda$-Lipschitz gradient with respect to the standard Euclidean norm $\|\cdot\|_2$. Then function $F(\cdot)$ satisfies Assumption (A2) with the matrix spectral norm, $\|\cdot\| = \|\cdot\|_{\mathrm{op}}$, and the gradient Lipschitz constant*

$$L = \lambda \cdot \frac{1}{N} \sum_{i=1}^{N} \|a_i\|_2^2. \tag{38}$$

**Lemma 2.** *Let each function $\mathcal{L}_i(\cdot)$ in problem (37) be twice continuously differentiable and have a $\lambda$-Lipschitz Hessian with respect to the standard Euclidean norm $\|\cdot\|_2$. Then function $F(\cdot)$ satisfies Assumption (A5) with the matrix spectral norm, $\|\cdot\| = \|\cdot\|_{\mathrm{op}}$, and the Hessian Lipschitz constant*

$$H = \lambda \cdot \frac{1}{N} \sum_{i=1}^{N} \|a_i\|_2^3. \tag{39}$$

## A.2    COMPARISON WITH PETHICK ET AL. (2025)

In this section, we provide a brief comparison of our results with the concurrent work of Pethick et al. (2025). We discovered this work only after the first version of our paper appeared online. In contrast to our trust-region approach, Pethick et al. (2025) build their analysis on the stochastic conditional gradient method of Mokhtari et al. (2020). Both these approaches are closely related, and we leave further investigation of this connection for future work. Besides, they provide analysis for arbitrary non-Euclidean norms, just as we do. However, their theoretical results have the following disadvantages:

(i) The analysis of Pethick et al. (2025) is limited to the case where the regularizer $R(\cdot)$ is the indicator function of some closed and convex set. In contrast, our analysis works with an arbitrary proper, closed, and convex regularizer.

(ii) Pethick et al. (2025) obtain the same $1/\varepsilon^4$ complexity for Muon, both with and without weight decay. In particular, they do not obtain the improved $1/\varepsilon^3$ complexity under the star-convexity assumption. Besides, their theoretical results provide a limited explanation for the success of Muon with weight decay in the training of large-scale language models (Liu et al., 2025), and for the success of Muon without weight decay in the training of small-scale language models (Jordan et al., 2024). In contrast, we discuss both cases in Section 4.2 and appendix C.

**(iii)** In the case of non-convex functions, Pethick et al. (2025) establish only the complexity $1/\varepsilon^4$, which is standard for non-convex functions. In contrast, we obtain the improved $1/\varepsilon^{3.5}$ complexity by utilizing an additional extrapolation step and the second-order smoothness assumption.

## A.3 COMPLEXITIES FOR NON-CONVEX AND STAR-CONVEX FUNCTIONS

In this paper, we often compare the complexities of finding an $\varepsilon$-accurate solution to problem (1) with non-convex and star-convex objective functions $f(x)$. It is important to highlight that we use the approximate first-order stationarity $\|\nabla f(x_k) + \hat{\nabla} R_k\|_* \leq \varepsilon$ for non-convex functions, and the approximate functional suboptimality $F(x_k) - F(x^*) \leq \varepsilon$. For instance, in Section 3.2, comparing the $1/\varepsilon^3$ complexity result in Corollary 4 for star-convex functions with the $1/\varepsilon^{3.5}$ result of Cutkosky & Mehta (2020); Sun et al. (2023) for non-convex functions may seem incorrect.

However, by slightly modifying the proof of Theorem 8, we can obtain the same improved complexity $1/\varepsilon^3$ for finding an $\varepsilon$-stationary point under the star-convexity Assumption (A4) and bounded domain Assumption (A6). This can be correctly compared with the results for non-convex functions, such as Corollaries 2 and 5. On the other hand, under these assumptions, we can upper-bound the functional suboptimality as follows:

$$F(x_k) - F(x^*) \leq D\|\nabla f(x_k) + \hat{\nabla} R_k\|_*. \tag{40}$$

Hence, the generalized $(\varepsilon/D)$-stationarity $\|\nabla f(x_k) + \hat{\nabla} R_k\|_*$ implies the $\varepsilon$-approximate functional suboptimality as well. Note that, strictly speaking, these considerations are not valid for the case of zero regularization, $R \equiv 0$. We leave further investigation for future work.

## A.4 PROOF OF LEMMA 1

The inequality in Assumption (A2) is implied by the following inequality, which we further prove:

$$\nabla F(\mathbf{X}_1)[\mathbf{H}] - \nabla F(\mathbf{X}_2)[\mathbf{H}] \leq L\|\mathbf{X}_1 - \mathbf{X}_2\|_{\mathrm{op}}\|\mathbf{H}\|_{\mathrm{op}}, \tag{41}$$

where $\mathbf{X}_1, \mathbf{H} \in \mathbb{R}^{m \times n}$. We proceed as follows:

$$\begin{aligned}
\nabla F(\mathbf{X}_1)[\mathbf{H}] &- \nabla F(\mathbf{X}_2)[\mathbf{H}] \\
&\overset{(a)}{\leq} \frac{1}{N} \sum_{i=1}^{N} \left( \nabla \mathcal{L}_i(\mathbf{X}_1 a_i)[\mathbf{H} a_i] - \nabla \mathcal{L}_i(\mathbf{X}_2 a_i)[\mathbf{H} a_i] \right) \\
&\overset{(b)}{\leq} \frac{1}{N} \sum_{i=1}^{N} \lambda \|\mathbf{X}_1 a_i - \mathbf{X}_2 a_i\|_2 \|\mathbf{H} a_i\|_2 \\
&\overset{(c)}{\leq} \frac{1}{N} \sum_{i=1}^{N} \lambda \|\mathbf{X}_1 - \mathbf{X}_2\|_{\mathrm{op}} \|\mathbf{H}\|_{\mathrm{op}} \|a_i\|_2^2 \\
&\overset{(d)}{=} L\|\mathbf{X}_1 - \mathbf{X}_2\|_{\mathrm{op}} \|\mathbf{H}\|_{\mathrm{op}},
\end{aligned}$$

where (a) uses the linearity of the first differential; (b) uses the $\lambda$-Lipschitzness of the gradients $\nabla \mathcal{L}_i$ with respect to the standard Euclidean norm $\|\cdot\|_2$; (c) uses the properties of the matrix spectral norm $\|\cdot\|_{\mathrm{op}}$; (d) uses the definition of $L$ in eq. (38). $\qquad\square$

## A.5 PROOF OF LEMMA 2

The inequality in Assumption (A5) is implied by the following inequality, which we further prove:

$$\nabla^2 F(\mathbf{X}_1)[\mathbf{X}_1 - \mathbf{X}_2, \mathbf{H}] - \nabla^2 F(\mathbf{X}_2)[\mathbf{X}_1 - \mathbf{X}_2, \mathbf{H}] \leq H\|\mathbf{X}_1 - \mathbf{X}_2\|_{\mathrm{op}}^2 \|\mathbf{H}\|_{\mathrm{op}}, \tag{42}$$

where $\mathbf{X}_1, \mathbf{X}_2, \mathbf{H} \in \mathbb{R}^{m \times n}$. We proceed as follows:

$$\nabla^2 F(\mathbf{X}_1)[\mathbf{X}_1 - \mathbf{X}_2, \mathbf{H}] - \nabla^2 F(\mathbf{X}_2)[\mathbf{X}_1 - \mathbf{X}_2, \mathbf{H}]$$

$$\overset{(a)}{\leq} \frac{1}{N} \sum_{i=1}^{N} \left( \nabla^2 \mathcal{L}_i(\mathbf{X}_1 a_i)[(\mathbf{X}_1 - \mathbf{X}_2)a_i, \mathbf{H}a_i] - \nabla^2 \mathcal{L}_i(\mathbf{X}_2 a_i)[(\mathbf{X}_1 - \mathbf{X}_2)a_i, \mathbf{H}a_i] \right)$$

$$\overset{(b)}{\leq} \frac{1}{N} \sum_{i=1}^{N} \lambda \|\mathbf{X}_1 a_i - \mathbf{X}_2 a_i\|_2 \|(\mathbf{X}_1 - \mathbf{X}_2)a_i\|_2 \|\mathbf{H}a_i\|_2$$

$$\overset{(c)}{\leq} \frac{1}{N} \sum_{i=1}^{N} \lambda \|\mathbf{X}_1 - \mathbf{X}_2\|_{\mathrm{op}}^2 \|\mathbf{H}\|_{\mathrm{op}} \|a_i\|_2^3$$

$$\overset{(d)}{=} H \|\mathbf{X}_1 - \mathbf{X}_2\|_{\mathrm{op}}^2 \|\mathbf{H}\|_{\mathrm{op}},$$

where (a) uses the multilinearity of the second differential; (b) uses the $\lambda$-Lipschitzness of the Hessians $\nabla^2 \mathcal{L}_i$ with respect to the standard Euclidean norm $\|\cdot\|_2$; (c) uses the properties of the matrix spectral norm $\|\cdot\|_{\mathrm{op}}$; (d) uses the definition of $H$ in eq. (39). $\qquad \square$

# B  MUON VS ORTHOGONAL-SGDM

As previously discussed in Section 3.1, one can verify that in the case of the non-Euclidean spectral norm $\|\cdot\| = \|\cdot\|_{\mathrm{op}}$ and zero regularizer $R \equiv 0$, Algorithm 1 exactly matches the Muon optimizer as it was originally described by Jordan et al. (2024). As mentioned in Section 1.1, there is another variant of the orthogonalized gradient method with momentum proposed by Tuddenham et al. (2022), namely, Orthogonal-SGDM. We provide pseudocode for both algorithms in Algorithm 4. While Muon can achieve impressive practical results (Jordan et al., 2024; Liu et al., 2025), Orthogonal-SGDM is unable to outperform Muon or well-tuned standard SGD with momentum (Tuddenham et al., 2022; Jordan et al., 2024). In this section, we provide a possible explanation for this phenomenon using our theoretical insights from Section 3.2.

---

**Algorithm 4** Muon/Orthogonal-SGDM (Jordan et al., 2024; Tuddenham et al., 2022)

---

1: **input:** $\mathbf{X}_0, \mathbf{M}_0 \in \mathbb{R}^{m \times n}$
2: **parameters:** $\eta > 0$, $\alpha \in (0, 1)$, $K \in \{1, 2, \dots\}$
3: **for** $k = 0, 1, \dots, K - 1$ **do**
4: $\quad$ Compute $\mathbf{G}_k$, such that $\mathbb{E}[\mathbf{G}_k] = \nabla F(\mathbf{X}_k)$
5: $\quad$ Compute $\mathbf{X}_{k+1}$ and $\mathbf{M}_{k+1}$ as follows:

$\qquad \mathbf{X}_{k+1} = \mathbf{X}_k - \eta \mathbf{O}_{k+1}, \ \mathbf{M}_{k+1} = (1 - \alpha)\mathbf{M}_k + \alpha \mathbf{G}_k, \quad \mathbf{O}_{k+1} = \mathrm{orth}(\mathbf{M}_{k+1})$ (Muon)
$\qquad \mathbf{X}_{k+1} = \mathbf{X}_k - \eta \mathbf{M}_{k+1}, \ \mathbf{M}_{k+1} = (1 - \alpha)\mathbf{M}_k + \alpha \mathbf{O}_{k+1}, \ \mathbf{O}_{k+1} = \mathrm{orth}(\mathbf{G}_k)$ (OSGDM)

6: **output:** $\mathbf{X}_K \in \mathbb{R}^{m \times n}$

---

The main difference between the updates (Muon) and (OSGDM) is the order in which the momentum and orthogonalization are applied. Our theory, in particular Lemmas 4 and 5 in Appendix D.2, suggests that the update rule (Muon) is preferable:

**(i)** First, by analyzing the proof of Lemma 5, one can conclude that the momentum accumulates a weighted sum of the terms $\zeta_k = \nabla f(x_k) - g(x_k, \xi_k)$. On the other hand, from Assumption (A1), it follows that the variance of $\zeta_k$ is bounded. What is even more important, this assumption implies that the random variables $\zeta_k$ are independent and centered, i.e., $\mathbb{E}[\zeta_k] = 0$, which means that the total variance of their weighted sum is reduced due to the so-called *batching effect*. Refer to the proof of Lemma 5 in Appendix D.2.2 for details.

**(ii)** Second, recall our trust-region interpretation of the orthogonalized gradient descent. According to the discussion in Section 3.1, it is natural to feed the momentum term $m_{k+1}$ to the trust-region optimization step (15) with the non-Euclidean spectral norm, which results in the ¡¡orthogonalized momentum¿¿ $\mathbf{O}_{k+1}$ in eq. (Muon) rather than the ¡¡momentum of orthogonals¿¿ $\mathbf{M}_{k+1}$ in eq. (OSGDM). These considerations are supported by the proof of Lemma 4 in Appendix D.2.1.

In contrast, the considerations above are not applicable to the update rule (OSGDM). Indeed, there is no reason to believe that the terms $\mathbf{O}_{k+1}$ in eq. (OSGDM) are unbiased in any sense or that their accumulated variance can be reduced. Additionally, it is unclear what the possible interpretation of the momentum term $\mathbf{M}_{k+1}$, constructed from the orthogonalized parts $\mathbf{O}_{k+1}$ in eq. (OSGDM), could be.

# C   Algorithms with Weight Clipping for Star-Convex Functions

As mentioned in Section 4.1, it is possible to obtain improved convergence guarantees for the iterations (9) under the star-convexity Assumption (A4), provided that these iterations are bounded. Recall that the iterations (19) and Algorithm 2 use weight decay to achieve this. An alternative approach is to simply assume the boundedness of $\mathrm{dom}\, R$, which implies the following inequality:

$$\|x - x'\| \leq D \quad \text{for all } x, x' \in \mathrm{dom}\, R, \tag{A6}$$

where $D > 0$ is the diameter of $\mathrm{dom}\, R$. In this section, we provide convergence guarantees for the iterations (9) and Algorithm 1 for star-convex functions under the additional Assumption (A6).

We start with the result in Theorem 7 for the deterministic iterations (9). The proof is available in Appendix D.7. Next, we obtain the convergence result for the stochastic Algorithm 1 in Theorem 8. The proof is available in Appendix D.8. Finally, using Theorems 7 and 8, it is not hard to obtain the explicit iteration complexities for the iterations (9) and Algorithm 1 in Corollaries 7 and 8, respectively. The proofs are omitted due to their simplicity.

**Theorem 7.** *Let Assumptions (A2), (A4) and (A6) hold, and let $x_0 \in \mathrm{dom}\, R$. Then the iterations* (9) *satisfy the following inequality:*

$$F(x_K) - F(x^*) \leq \left(\frac{D}{\eta + D}\right)^K (F(x_0) - F(x^*)) + \frac{3LD\eta}{2}. \tag{43}$$

**Theorem 8.** *Let Assumptions (A1) to (A4) and (A6) hold, and let $x_0 \in \mathrm{dom}\, R$ and $m_0 = g(x_0, \xi_0)$. Then the iterations of Algorithm 1 satisfy the following inequality:*

$$\mathbb{E}\left[F(x_K) - F(x^*)\right] \leq \left(\frac{D}{\eta + D}\right)^K (F(x_0) - F(x^*)) + 2\sqrt{\alpha}D\rho\sigma + \frac{2\eta\rho\sigma}{\alpha} + \frac{3LD\eta}{2} + \frac{2LD\eta}{\alpha}. \tag{44}$$

**Corollary 7.** *Under the conditions of Theorem 7, to reach the precision $F(x_K) - F(x^*) \leq \varepsilon$ by the iterations* (9), *it is sufficient to choose the stepsize $\eta$ and the number of iterations $K$ as follows:*

$$\eta = \mathcal{O}\left(\frac{\varepsilon}{LD}\right), \qquad K = \tilde{\mathcal{O}}\left(\max\left\{1, \frac{LD^2}{\varepsilon}\right\}\right). \tag{45}$$

**Corollary 8.** *Under the conditions of Theorem 8, to reach the precision $\mathbb{E}\left[F(x_K) - F(x^*)\right] \leq \varepsilon$ by Algorithm 1, it is sufficient to choose the parameters as follows:*

$$\eta = \mathcal{O}\left(\min\left\{\frac{\varepsilon}{LD}, \frac{\varepsilon}{\rho\sigma}, \frac{\varepsilon^3}{D^2\rho^3\sigma^3}, \frac{\varepsilon^3}{LD^3\rho^2\sigma^2}\right\}\right), \qquad \alpha = \mathcal{O}\left(\min\left\{1, \frac{\varepsilon^2}{D^2\rho^2\sigma^2}\right\}\right), \tag{46}$$

$$K = \tilde{\mathcal{O}}\left(\max\left\{1, \frac{LD^2}{\varepsilon}, \frac{D\rho\sigma}{\varepsilon}, \frac{D^3\rho^3\sigma^3}{\varepsilon^3}, \frac{LD^4\rho^2\sigma^2}{\varepsilon^3}\right\}\right). \tag{47}$$

The complexity of the iterations (9) without weight decay in Corollary 7 matches the complexity of the iterations (6) with weight decay in Corollary 3. Similarly, the complexity of Algorithm 1 without weight decay in Corollary 8 matches the result for Algorithm 2 with weight decay in Corollary 4. It is important to highlight that Assumption (A6) obviously does not hold in the case of zero regularization $R \equiv 0$. To tackle this issue, one could choose the regularizer $R(\cdot)$ as the following indicator function:

$$R(x) = \begin{cases} 0 & \|x\| \leq D \\ +\infty & \|x\| > D \end{cases}, \quad \text{where } D \geq \|x^*\|. \tag{48}$$

In the case of the infinity norm $\|\cdot\| = \|\cdot\|_\infty$, Algorithm 1 with the regularizer (48) can be easily implemented and reduces to signSGD with momentum and weight clipping. Weight clipping is used in various machine learning applications (Elsayed et al., 2024; Bernstein et al., 2020; Arjovsky et al., 2017) and can be seen as an alternative to weight decay used in Algorithm 2.

Unfortunately, in the case of the spectral norm $\|\cdot\| = \|\cdot\|_{\mathrm{op}}$, Algorithm 1 with the regularizer (48), such a clipping procedure is not easily implementable, which justifies the use of weight decay in Muon. On the other hand, Liu et al. (2025) reported that in cases where language models are small-scale and/or the training time is not lengthy, the iterations of Muon do not grow excessively large. Hence, with a reasonable choice of the radius $D$, the weight clipping step in Algorithm 1 would be skipped. Thus, the fact that we obtain the same complexity results for Algorithm 1 with weight clipping and Algorithm 2 weight decay may support the practical evidence that weight decay is not necessary for the training of small-scale language models (Liu et al., 2025; Jordan et al., 2024).

## D    PROOFS OF THEOREMS

### D.1    PROOF OF THEOREM 1

We start with the following Lemma 3, which allows us to incorporate an arbitrary proper, closed, and convex regularizer $R(x)$ into the non-Euclidean trust-region optimization step. The proof is available in Appendix D.1.1.

**Lemma 3.** *Let $z_+ \in \mathcal{X}$ be defined as a solution to the following trust-region optimization problem:*

$$z_+ = \underset{x \in \mathcal{X}}{\arg\min} \ \Psi(x) + R(x) \quad s.t. \quad \|x - z\| \leq \eta, \tag{49}$$

*where $\Psi(\cdot) \colon \mathcal{X} \to \mathbb{R}$ is a differentiable function, $z \in \mathrm{dom}\, R$ is a center of the trust-region, and $\eta > 0$ is the trust-region radius. Then the following inequality holds:*

$$R(z) + \langle \nabla \Psi(z_+), z - z_+ \rangle \geq R(z_+) + \eta \|\nabla \Psi(z_+) + \hat{\nabla} R\|_*, \tag{50}$$

*where $\hat{\nabla} R \in \partial R(z_+)$.*

Now, we are ready to prove Theorem 1. We can upper-bound $F(x_{k+1})$ as follows:

$$
\begin{aligned}
F(x_{k+1}) &\overset{(a)}{=} f(x_{k+1}) + R(x_{k+1}) \\
&\overset{(b)}{\leq} f(x_k) + \langle \nabla f(x_k), x_{k+1} - x_k \rangle + \tfrac{1}{2} L \|x_{k+1} - x_k\|^2 + R(x_{k+1}) \\
&\overset{(c)}{\leq} f(x_k) + R(x_k) - \eta \|\nabla f(x_k) + \hat{\nabla} R_{k+1}\|_* + \tfrac{1}{2} L \|x_{k+1} - x_k\|^2 \\
&\overset{(d)}{=} f(x_k) + R(x_k) - \eta \|\nabla f(x_{k+1}) + \hat{\nabla} R_{k+1}\|_* + \eta \|\nabla f(x_{k+1}) - \nabla f(x_k)\|_* \\
&\quad + \tfrac{1}{2} L \|x_{k+1} - x_k\|^2 \\
&\overset{(e)}{\leq} f(x_k) + R(x_k) - \eta \|\nabla f(x_{k+1}) + \hat{\nabla} R_{k+1}\|_* + L\eta \|x_{k+1} - x_k\| + \tfrac{1}{2} L \|x_{k+1} - x_k\|^2 \\
&\overset{(f)}{\leq} f(x_k) + R(x_k) - \eta \|\nabla f(x_{k+1}) + \hat{\nabla} R_{k+1}\|_* + \tfrac{3}{2} L \eta^2 \\
&\overset{(g)}{=} F(x_k) - \eta \|\nabla f(x_{k+1}) + \hat{\nabla} R_{k+1}\|_* + \tfrac{3}{2} L \eta^2,
\end{aligned}
$$

where (a) and (g) use the definition of $F(x)$ in problem (1); (b) and (e) use Assumption (A2); (c) uses Lemma 3 with $\Psi(x) = \langle \nabla f(x_k), x \rangle$ and $(z, z_+) = (x_k, x_{k+1})$; (d) uses the triangle inequality; (f) uses the fact that $\|x_{k+1} - x_k\| \leq \eta$, which is implied by eq. (9).    $\square$

### D.1.1    PROOF OF LEMMA 3

The objective function in problem (49) is lower semi-continuous, because function $R(x)$ is proper, closed and convex, and the constrained set in this problem is obviously compact. Hence, there exists at least a single solution $z^+ \in \mathcal{X}$ to the problem.

Let function $I_\eta(x)$ be the indicator function of the constraint set in problem (49), i.e.,

$$I_\eta(x) = \begin{cases} 0 & \|x - z\| \leq \eta \\ +\infty & \|x - z\| > \eta \end{cases}, \tag{51}$$

and let function $R_\eta(x)$ be defined as follows:

$$R_\eta(x) = R(x) + I_\eta(x). \tag{52}$$

It is easy to verify that function $R_\eta(x)$ is proper, closed, and convex, and problem (49) is equivalent to the following:

$$\min_{x \in \mathcal{X}} \Psi(x) + R_\eta(x). \tag{53}$$

Therefore, $z_+$ satisfies the following inequality for all $x \in \mathcal{X}$:

$$\Psi(x) + R_\eta(x) \geq \Psi(z_+) + R_\eta(z_+). \tag{54}$$

Our next goal is to show that $-\nabla\Psi(z_+) \in \partial R_\eta(z_+)$, which we prove by contradiction. Suppose that the opposite holds, i.e., $-\nabla\Psi(z_+) \notin \partial R_\eta(z_+)$. This implies the existence of $x \in \mathcal{X}$ and $\delta > 0$ such that

$$R_\eta(x) - R_\eta(z_+) + \langle \nabla\Psi(z_+), x - z_+ \rangle = -\delta. \tag{55}$$

On the other hand, by the definition of the differentiability, we have

$$\lim_{\tau \to 0} \frac{\Psi(z_+ + \tau(x - z_+)) - \Psi(z_+) - \tau\langle\nabla\Psi(z_+), x - z_+\rangle}{\tau} = 0. \tag{56}$$

Hence, there exists $\tau \in (0, 1)$ such that

$$\Psi(z_+ + \tau(x - z_+)) - \Psi(z_+) - \tau\langle\nabla\Psi(z_+), x - z_+\rangle \leq \tfrac{1}{2}\delta\tau. \tag{57}$$

Hence, we obtain the following:

$$
\begin{aligned}
-\delta &\overset{(a)}{=} R_\eta(x) - R_\eta(z_+) + \langle\nabla\Psi(z_+), x - z_+\rangle \\
&\overset{(b)}{\geq} \frac{1}{\tau}\big(R_\eta(z_+ + \tau(x - z_+)) - R_\eta(z_+) + \tau\langle\nabla\Psi(z_+), x - z_+\rangle\big) \\
&\overset{(c)}{\geq} -\frac{1}{\tau}\big(\Psi(z_+ + \tau(x - z_+)) - \Psi(z_+) - \tau\langle\nabla\Psi(z_+), x - z_+\rangle\big) \\
&\overset{(d)}{\geq} -\tfrac{1}{2}\delta,
\end{aligned}
$$

where (a) uses eq. (55); (b) uses the convexity of function $R_\eta(x)$; (c) uses eq. (54); (d) uses eq. (57). This implies the contradiction $\delta \leq 0$. Thus, we have shown that $-\nabla\Psi(z_+) \in \partial R_\eta(z_+)$.

Our next goal is to show that $\operatorname{ri}\operatorname{dom} R \cap \operatorname{ri}\operatorname{dom} I_\eta \neq \varnothing$. In the case where $z \in \operatorname{ri}\operatorname{dom} R$, this statement is obviously true because $z \in \operatorname{ri}\operatorname{dom} I_\eta$. Hence, it remains to consider the case $z \notin \operatorname{ri}\operatorname{dom} R$. By Theorem 6.3 of Rockafellar (1997), we have $\operatorname{cl}(\operatorname{ri}\operatorname{dom} R) = \operatorname{cl}\operatorname{dom} R \supset \operatorname{dom} R$. Hence, there exists a sequence of vectors $\{x_k\}_{k=0}^\infty \subset \operatorname{ri}\operatorname{dom} R$ such that $\lim_{k\to\infty}\|x_k - z\| = 0$. Since $I_\eta(x)$ is the indicator of the ball with the center at $z$, we can find $x_k \in \operatorname{ri}\operatorname{dom} I_\eta$. Thus, we have $x_k \in \operatorname{ri}\operatorname{dom} I_\eta \cap \operatorname{ri}\operatorname{dom} R \neq \varnothing$.

Using the definition of $R_\eta(x)$ in eq. (52), the fact that $\operatorname{ri}\operatorname{dom} R \cap \operatorname{ri}\operatorname{dom} I_\eta \neq \varnothing$, and Theorem 23.8 of Rockafellar (1997), we have $\partial R_\eta(z_+) = \partial R(z_+) + \partial I_\eta(z_+)$. Hence, using the fact that $-\nabla\Psi(z_+) \in \partial R_\eta(z_+)$, we have

$$\nabla\Psi(z_+) + \hat{\nabla}R + \hat{\nabla}I_\eta = 0, \tag{58}$$

where $\hat{\nabla}R \in \partial R(z_+)$, and $\hat{\nabla}I_\eta \in \partial I_\eta(z_+)$. Therefore, we obtain the following:

$$
\begin{aligned}
R(z) + \langle\nabla\Psi(z_+), z - z_+\rangle &\overset{(a)}{\geq} R(z_+) + \langle\nabla\Psi(z_+) + \hat{\nabla}R, z - z_+\rangle \\
&\overset{(b)}{=} R(z_+) - \langle\hat{\nabla}I_\eta, z - z_+\rangle \\
&= R(z_+) + \langle\hat{\nabla}I_\eta, z_+\rangle - \langle\hat{\nabla}I_\eta, z\rangle \\
&\overset{(c)}{=} R(z_+) + I_\eta(z_+) + I_\eta^*(\hat{\nabla}I_\eta) - \langle\hat{\nabla}I_\eta, z\rangle \\
&\overset{(d)}{=} R(z_+) + \eta\|\hat{\nabla}I_\eta\|_* \\
&\overset{(e)}{=} R(z_+) + \eta\|\nabla\Psi(z_+) + \hat{\nabla}R\|_*,
\end{aligned}
$$

where (a) uses the convexity of function $R(z)$; (b) and (e) use eq. (58); (c) uses the Fenchel-Young equality; (d) uses the definition of $I_\eta(x)$ and standard calculations. $\qquad\square$

### D.2 Proof of Theorem 2

We start with the following Lemma 4, which describes the update rule (15). It can be seen as an adaptation of the famous *descent lemma* in the context of trust-region optimization. The proof is, in many ways, based on Lemma 3 and is available in Appendix D.2.1.

**Lemma 4.** *Let Assumption (A2) hold, and let $x_0 \in \operatorname{dom} R$. Then the iterations of Algorithm 1 satisfy the following inequality:*

$$F(x_{k+1}) \leq F(x_k) - \eta \|\nabla f(x_{k+1}) + \hat{\nabla} R_{k+1}\|_* + 2\eta \|\nabla f(x_{k+1}) - m_{k+1}\|_* + \tfrac{3}{2} L \eta^2, \quad (59)$$

*where $\hat{\nabla} R_{k+1} \in \partial R(x_{k+1})$.*

The next Lemma 5 describes the dynamics of the momentum term, which is updated according to eq. (14). In particular, it upper-bounds the expected distance between the momentum $m_k$ and the true gradient $\nabla f(x_k)$. The proof can be seen as an adaptation of the proof by Cutkosky & Mehta (2020) to the non-Euclidean norm setting and is available in Appendix D.2.2.

**Lemma 5.** *Let Assumptions (A1) to (A3) hold, and let $x_0 \in \operatorname{dom} R$ and $m_0 = g(x_0, \xi_0)$. Then the iterations of Algorithm 1 satisfy the following inequality for $k \geq 0$:*

$$\mathbb{E}\left[\|m_{k+1} - \nabla f(x_k)\|_*\right] \leq (1-\alpha)^{k+1} \rho\sigma + \sqrt{\alpha}\rho\sigma + \frac{L\eta}{\alpha}. \quad (60)$$

Now, we are ready to prove Theorem 2. Using Lemma 4, we obtain the following inequality:

$$\min_{k=1,\dots,K} \|\nabla f(x_k) + \hat{\nabla} R_k\|_* \leq \frac{F(x_0) - \inf_x F(x)}{\eta K} + \frac{3L\eta}{2} + \frac{2}{K}\sum_{k=1}^{K}\|\nabla f(x_k) - m_k\|_*$$

$$\overset{(a)}{\leq} \frac{F(x_0) - \inf_x F(x)}{\eta K} + \frac{7L\eta}{2} + \frac{2}{K}\sum_{k=0}^{K-1}\|\nabla f(x_k) - m_{k+1}\|_*,$$

where (a) uses the triangle inequality, Assumption (A2), and eq. (15). Using Lemma 5, we obtain

$$\mathbb{E}\left[\min_{k=1,\dots,K}\|\nabla f(x_k) + \hat{\nabla} R_k\|_*\right] \leq \frac{F(x_0) - \inf_x F(x)}{\eta K} + \frac{7L\eta}{2} + \frac{2L\eta}{\alpha} + \frac{2\rho\sigma}{\alpha K} + 2\sqrt{\alpha}\rho\sigma.$$

$$\square$$

#### D.2.1 Proof of Lemma 4

We can upper-bound $F(x_{k+1})$ as follows:

$$
\begin{aligned}
F(x_{k+1}) &\overset{(a)}{=} f(x_{k+1}) + R(x_{k+1})\\
&\overset{(b)}{\leq} f(x_k) + \langle \nabla f(x_k), x_{k+1} - x_k\rangle + \tfrac{1}{2}L\|x_{k+1} - x_k\|^2 + R(x_{k+1})\\
&= f(x_k) + \tfrac{1}{2}L\|x_{k+1} - x_k\|^2 + R(x_{k+1})\\
&\quad + \langle m_{k+1} + \nabla f(x_{k+1}) - m_{k+1} + \nabla f(x_k) - \nabla f(x_{k+1}), x_{k+1} - x_k\rangle\\
&\overset{(c)}{\leq} f(x_k) + \tfrac{1}{2}L\|x_{k+1} - x_k\|^2 + R(x_{k+1}) + \langle m_{k+1}, x_{k+1} - x_k\rangle\\
&\quad + \|x_{k+1} - x_k\|\|\nabla f(x_{k+1}) - m_{k+1}\|_* + \|x_{k+1} - x_k\|\|\nabla f(x_k) - \nabla f(x_{k+1})\|_*\\
&\overset{(d)}{\leq} f(x_k) + \tfrac{3}{2}L\|x_{k+1} - x_k\|^2 + \|x_{k+1} - x_k\|\|\nabla f(x_{k+1}) - m_{k+1}\|_*\\
&\quad + R(x_{k+1}) + \langle m_{k+1}, x_{k+1} - x_k\rangle\\
&\overset{(e)}{\leq} f(x_k) + \tfrac{3}{2}L\eta^2 + \eta\|\nabla f(x_{k+1}) - m_{k+1}\|_* + R(x_{k+1}) + \langle m_{k+1}, x_{k+1} - x_k\rangle\\
&\overset{(f)}{\leq} f(x_k) + \tfrac{3}{2}L\eta^2 + \eta\|\nabla f(x_{k+1}) - m_{k+1}\|_* + R(x_k) - \eta\|m_{k+1} + \hat{\nabla} R_{k+1}\|_*\\
&\overset{(g)}{=} F(x_k) + \tfrac{3}{2}L\eta^2 + \eta\|\nabla f(x_{k+1}) - m_{k+1}\|_* - \eta\|m_{k+1} + \hat{\nabla} R_{k+1}\|_*\\
&\overset{(h)}{\leq} F(x_k) + \tfrac{3}{2}L\eta^2 + 2\eta\|\nabla f(x_{k+1}) - m_{k+1}\|_* - \eta\|\nabla f(x_{k+1}) + \hat{\nabla} R_{k+1}\|_*,
\end{aligned}
$$

where (a) and (g) use the definition of function $F(x)$ in problem (1); (b) and (d) use Assumption (A2); (c) uses the definition of the dual norm; (e) uses the fact that $\|x_{k+1} - x_k\| \leq \eta$, which is implied by eq. (15); (f) uses eq. (15) and Lemma 3 with $\Psi(x) = \langle m_{k+1}, x \rangle$ and $(z_+, z) = (x_{k+1}, x_k)$; (h) uses the triangle inequality. □

### D.2.2 PROOF OF LEMMA 5

We can express $m_{k+1} - \nabla f(x_k)$ as follows using eq. (14):

$$
\begin{aligned}
m_{k+1} - \nabla f(x_k) &= (1 - \alpha)m_k + \alpha g(x_k; \xi_k) - \nabla f(x_k) \\
&= (1 - \alpha)(m_k - \nabla f(x_{k-1})) + \alpha(g(x_k; \xi_k) - \nabla f(x_k)) \\
&\quad + (1 - \alpha)(\nabla f(x_{k-1}) - \nabla f(x_k)).
\end{aligned}
$$

This implies the following for all $k \geq 0$:

$$
m_{k+1} - \nabla f(x_k) = (1 - \alpha)^{k+1}(m_0 - \nabla f(x_0)) + \sum_{i=0}^{k-1} (1 - \alpha)^{k-i}(\nabla f(x_i) - \nabla f(x_{i+1}))
$$

$$
+ \sum_{i=0}^{k} \alpha(1 - \alpha)^{k-i}(g(x_i, \xi_i) - \nabla f(x_i)).
$$

Using this, we can upper-bound $\mathbb{E}\left[\|m_{k+1} - \nabla f(x_k)\|_*\right]$ for $k \geq 0$ as follows:

$$
\mathbb{E}\left[\|m_{k+1} - \nabla f(x_k)\|_*\right] \overset{(a)}{\leq} (1 - \alpha)^{k+1}\mathbb{E}\left[\|m_0 - \nabla f(x_0)\|_*\right]
$$

$$
+ \sum_{i=0}^{k-1} (1 - \alpha)^{k-i}\|\nabla f(x_i) - \nabla f(x_{i+1})\|_*
$$

$$
+ \mathbb{E}\left[\|\sum_{i=0}^{k} \alpha(1 - \alpha)^{k-i}(g(x_i, \xi_i) - \nabla f(x_i))\|_*\right]
$$

$$
\overset{(b)}{\leq} (1 - \alpha)^{k+1}\mathbb{E}\left[\|m_0 - \nabla f(x_0)\|_*\right] + \sum_{i=0}^{k-1} (1 - \alpha)^{k-i}L\eta
$$

$$
+ \mathbb{E}\left[\|\sum_{i=0}^{k} \alpha(1 - \alpha)^{k-i}(g(x_i, \xi_i) - \nabla f(x_i))\|_*\right]
$$

$$
\overset{(c)}{\leq} (1 - \alpha)^{k+1}\rho\mathbb{E}\left[\|m_0 - \nabla f(x_0)\|_2\right] + \sum_{i=0}^{k-1} (1 - \alpha)^{k-i}L\eta
$$

$$
+ \rho\mathbb{E}\left[\|\sum_{i=0}^{k} \alpha(1 - \alpha)^{k-i}(g(x_i, \xi_i) - \nabla f(x_i))\|_2\right]
$$

$$
\overset{(d)}{\leq} (1 - \alpha)^{k+1}\rho\sqrt{\mathbb{E}\left[\|m_0 - \nabla f(x_0)\|_2^2\right]} + \sum_{i=0}^{k-1} (1 - \alpha)^{k-i}L\eta
$$

$$
+ \rho\sqrt{\mathbb{E}\left[\|\sum_{i=0}^{k} \alpha(1 - \alpha)^{k-i}(g(x_i, \xi_i) - \nabla f(x_i))\|_2^2\right]}
$$

$$
\overset{(e)}{\leq} (1 - \alpha)^{k+1}\rho\sigma + \sum_{i=0}^{k-1} (1 - \alpha)^{k-i}L\eta + \alpha\rho\sigma\sqrt{\sum_{i=0}^{k} (1 - \alpha)^{2(k-i)}}
$$

$$
\leq (1 - \alpha)^{k+1}\rho\sigma + \frac{L\eta}{\alpha} + \sqrt{\alpha}\rho\sigma,
$$

where (a) uses the triangle inequality; (b) uses Assumption (A2) and eq. (15); (c) uses Assumption (A3); (d) uses Jensen's inequality; (e) uses Assumption (A1) and the fact that samples $\xi_i \sim \mathcal{D}$ are i.i.d.. □

### D.3 PROOF OF THEOREM 3

We start with the following Lemma 6. The proof is available in Appendix D.3.1.

**Lemma 6.** *Under the conditions of Theorem 3, let $x \in \mathcal{X}$ be defined as follows:*

$$x = \beta x^* + (1 - \beta)x_k. \tag{61}$$

*Then, the following inequalities hold:*

$$\|x - (1 - \beta)x_k\| \le \eta, \quad \|x - x_k\| \le 2\eta, \quad \|x - x_{k+1}\| \le 2\eta, \quad \|x_{k+1} - x_k\| \le 2\eta. \tag{62}$$

Now, we are ready to prove Theorem 3. We can upper-bound $F(x_{k+1})$ as follows:

$$
\begin{aligned}
F(x_{k+1}) &\overset{\text{(a)}}{=} f(x_{k+1}) + R(x_{k+1}) \\
&\overset{\text{(b)}}{\le} f(x_k) + \langle \nabla f(x_k), x_{k+1} - x_k \rangle + \tfrac{1}{2}L\|x_{k+1} - x_k\|^2 + R(x_{k+1}) \\
&\overset{\text{(c)}}{\le} f(x_k) + \langle \nabla f(x_k), x_{k+1} - x_k \rangle + R(x_{k+1}) + 2L\eta^2 \\
&\overset{\text{(d)}}{\le} f(x_k) + \langle \nabla f(x_k), x - x_k \rangle + R(x) + 2L\eta^2 \\
&\overset{\text{(e)}}{\le} f(x) + \tfrac{1}{2}L\|x - x_k\|^2 + R(x) + 2L\eta^2 \\
&\overset{\text{(f)}}{\le} f(x) + R(x) + 4L\eta^2 \\
&\overset{\text{(g)}}{\le} \beta(f(x^*) + R(x^*)) + (1 - \beta)(f(x_k) + R(x_k)) + 4L\eta^2 \\
&\overset{\text{(h)}}{=} \beta F(x^*) + (1 - \beta)F(x_k) + 4L\eta^2,
\end{aligned}
$$

where (a) and (h) use the definition of function $F(x)$ in problem (1); (b) and (e) use Assumption (A2); (c) and (f) use Lemma 6; (d) uses eq. (19) and the inequality $\|x - (1 - \beta)x_k\| \le \eta$, which is implied by Lemma 6; (g) uses assumption (A4) and the convexity if function $R(x)$. After rearranging, we obtain

$$F(x_{k+1}) - F(x^*) \le (1 - \beta)(F(x_k) - F(x^*)) + 4L\eta^2,$$

which implies the following inequality:

$$F(x_K) - F(x^*) \le (1 - \beta)^K (F(x_0) - F(x^*)) + \frac{4L\eta^2}{\beta}.$$

$\square$

#### D.3.1 PROOF OF LEMMA 6

First, we can show that $\beta\|x_k\| \le \eta$ by induction. Indeed $\beta\|x_0\| \le \eta$ due to eq. (22), and for all $k \in \{0, 1, \ldots, K - 1\}$, can obtain the following:

$$\beta\|x_{k+1}\| \overset{\text{(a)}}{\le} \beta\|x_{k+1} - (1 - \beta)x_k\| + (1 - \beta)\beta\|x_k\| \overset{\text{(b)}}{\le} \eta\beta + (1 - \beta)\eta \le \eta,$$

where (a) uses the triangle inequality; (b) uses eq. (19) and the induction hypothesis. Hence, we can prove the desired inequalities as follows:

$$\|x - (1 - \beta)x_k\| \overset{\text{(a)}}{=} \beta\|x^*\| \overset{\text{(b)}}{\le} \eta,$$

$$\|x - x_k\| \overset{\text{(c)}}{=} \beta\|x^* - x_k\| \overset{\text{(d)}}{\le} \beta\|x^*\| + \beta\|x_k\| \overset{\text{(e)}}{\le} 2\eta,$$

$$\|x - x_{k+1}\| \overset{\text{(f)}}{\le} \|x - (1 - \beta)x_k\| + \|x_{k+1} - (1 - \beta)x_k\| \overset{\text{(g)}}{\le} 2\eta,$$

$$\|x_{k+1} - x_k\| \overset{\text{(h)}}{\le} \|x_{k+1} - (1 - \beta)x_k\| + \beta\|x_k\| \overset{\text{(i)}}{\le} 2\eta,$$

where (a) and (c) use the definition of $x$ in eq. (61); (b) uses eq. (22); (d), (f) and (h) use the triangle inequality; (e) uses eq. (22) and the previously obtained inequality $\beta\|x_k\| \le \eta$; (g) uses eq. (19) and the previously obtained inequality $\|x - (1 - \beta)x_k\| \le \eta$; (i) uses eq. (19) and the previously obtained inequality $\beta\|x_k\| \le \eta$. $\square$

### D.4 PROOF OF THEOREM 4

In this proof, we are going to use Lemma 6, which is valid not only for the iterations (19), but also for Algorithm 2. We also obtain the following Lemma 7. The proof is almost identical to the proof of Lemma 5 in Appendix D.2.2, with the only difference being that we use the inequality $\|x_{k+1} - x_k\| \leq 2\eta$, which holds due to Lemma 6.

**Lemma 7.** *Let Assumptions (A1) to (A3) hold, and let $x_0 \in \operatorname{dom} R$ and $m_0 = g(x_0, \xi_0)$. Then the iterations of Algorithm 2 satisfy the following inequality for $k \geq 0$:*

$$\mathbb{E}\left[\|m_{k+1} - \nabla f(x_k)\|_*\right] \leq (1-\alpha)^{k+1}\rho\sigma + \sqrt{\alpha}\rho\sigma + \frac{2L\eta}{\alpha}. \tag{63}$$

Now, we are ready to prove Theorem 4. We can upper-bound $F(x_{k+1})$ as follows:

$$
\begin{aligned}
F(x_{k+1}) &\stackrel{(a)}{=} f(x_{k+1}) + R(x_{k+1}) \\
&\stackrel{(b)}{\leq} f(x_k) + \langle \nabla f(x_k), x_{k+1} - x_k \rangle + \tfrac{1}{2}L\|x_{k+1} - x_k\|^2 + R(x_{k+1}) \\
&\stackrel{(c)}{\leq} f(x_k) + \langle \nabla f(x_k) - m_{k+1}, x_{k+1} - x_k \rangle + 2L\eta^2 + \langle m_{k+1}, x_{k+1} - x_k \rangle + R(x_{k+1}) \\
&\stackrel{(d)}{\leq} f(x_k) + \langle \nabla f(x_k) - m_{k+1}, x_{k+1} - x_k \rangle + 2L\eta^2 + \langle m_{k+1}, x - x_k \rangle + R(x) \\
&= f(x_k) + \langle \nabla f(x_k), x - x_k \rangle + 2L\eta^2 + R(x) + \langle m_{k+1} - \nabla f(x_k), x - x_{k+1} \rangle \\
&\stackrel{(e)}{\leq} f(x_k) + \langle \nabla f(x_k), x - x_k \rangle + 2L\eta^2 + R(x) + \|x - x_{k+1}\|\|m_{k+1} - \nabla f(x_k)\|_* \\
&\stackrel{(f)}{\leq} f(x_k) + \langle \nabla f(x_k), x - x_k \rangle + 2L\eta^2 + R(x) + 2\eta\|m_{k+1} - \nabla f(x_k)\|_* \\
&\stackrel{(g)}{\leq} f(x) + \tfrac{1}{2}L\|x - x_k\|^2 + 2L\eta^2 + R(x) + 2\eta\|m_{k+1} - \nabla f(x_k)\|_* \\
&\stackrel{(h)}{\leq} f(x) + 4L\eta^2 + R(x) + 2\eta\|m_{k+1} - \nabla f(x_k)\|_* \\
&\stackrel{(i)}{\leq} \beta(f(x^*) + R(x^*)) + (1-\beta)(f(x_k) + R(x_k)) + 4L\eta^2 + 2\eta\|m_{k+1} - \nabla f(x_k)\|_* \\
&\stackrel{(j)}{=} \beta F(x^*) + (1-\beta)F(x_k) + 4L\eta^2 + 2\eta\|m_{k+1} - \nabla f(x_k)\|_*,
\end{aligned}
$$

where (a) and (j) use the definition of function $F(x)$ in problem (1); (b) and (g) use Assumption (A2); (c), (f) and (h) use Lemma 6; (d) uses eq. (21) and the inequality $\|x - (1-\beta)x_k\| \leq \eta$, which is implied by Lemma 6; (e) uses the definition of the dual norm; (i) uses assumption (A4) and the convexity if function $R(x)$. After rearranging, taking the expectation, and using Lemma 7, we get

$$
\begin{aligned}
\mathbb{E}\left[F(x_{k+1}) - F(x^*)\right] \leq{}& (1-\beta)\mathbb{E}\left[F(x_k) - F(x^*)\right] + 2\eta\rho\sigma(1-\alpha)^k + 2\eta\rho\sigma\sqrt{\alpha} \\
&+ 4L\eta^2 + \frac{4L\eta^2}{\alpha},
\end{aligned}
$$

which implies the following inequality:

$$\mathbb{E}\left[F(x_K) - F(x^*)\right] \leq (1-\beta)^K(F(x_0) - F(x^*)) + 2\eta\rho\sigma\left(\frac{1}{\alpha} + \frac{\sqrt{\alpha}}{\beta}\right) + \frac{4L\eta^2}{\beta}\left(1 + \frac{1}{\alpha}\right).$$

$\square$

### D.5    Proof of Theorem 5

In this proof, we are going to use Lemma 4, whis is valid for the iterations of Algorithm 3 as long as $\beta = 0$. We also obtain the following Lemma 8, which describes the dynamics of the momentum $m_k$ under the second-order smoothness. Similarly to the proofs of Lemma 5 and eq. (20), it can be seen as an adaptation of the proof by Cutkosky & Mehta (2020) to the non-Euclidean setting and is available in Appendix D.5.1.

**Lemma 8.** *Let Assumptions (A1), (A3) and (A5) hold, and let $x_0 \in \mathrm{dom}\, R$ and $m_0 = g(x_0, \xi_0)$. Then the iterations of Algorithm 3 satisfy the following inequality for $k \geq 0$:*

$$\mathbb{E}\left[\|m_{k+1} - \nabla f(x_k)\|_*\right] \leq (1-\alpha)^{k+1}\rho\sigma + \sqrt{\alpha}\rho\sigma + \frac{H\eta^2}{2\alpha^2}. \tag{64}$$

Now, we are ready to prove Theorem 5. Using Lemma 4, we obtain the following inequality:

$$\min_{k=1,\dots,K}\|\nabla f(x_k) + \hat{\nabla}R_k\|_* \leq \frac{F(x_0) - \inf_x F(x)}{\eta K} + \frac{3L\eta}{2} + \frac{2}{K}\sum_{k=1}^{K}\|\nabla f(x_k) - m_k\|_*$$

$$\overset{(a)}{\leq} \frac{F(x_0) - \inf_x F(x)}{\eta K} + \frac{7L\eta}{2} + \frac{2}{K}\sum_{k=0}^{K-1}\|\nabla f(x_k) - m_{k+1}\|_*,$$

where (a) uses the triangle inequality and Assumption (A2). Using Lemma 8, we obtain the following:

$$\mathbb{E}\left[\min_{k=1,\dots,K}\|\nabla f(x_k) + \hat{\nabla}R_k\|_*\right] \leq \frac{F(x_0) - \inf_x F(x)}{\eta K} + \frac{7L\eta}{2} + \frac{H\eta^2}{\alpha^2} + \frac{2\rho\sigma}{\alpha K} + 2\sqrt{\alpha}\rho\sigma.$$

$\square$

#### D.5.1    Proof of Lemma 8

We can express $m_{k+1} - \nabla f(x_k)$ as follows using eq. (28):

$$\begin{aligned}
m_{k+1} - \nabla f(x_k) &= (1-\alpha)m_k + \alpha g(\overline{x}_k; \xi_k) - \nabla f(x_k) \\
&= (1-\alpha)(m_k - \nabla f(x_{k-1})) + \alpha(g(\overline{x}_k; \xi_k) - \nabla f(\overline{x}_k)) \\
&\quad + \alpha\nabla f(\overline{x}_k) + (1-\alpha)\nabla f(x_{k-1}) - \nabla f(x_k)
\end{aligned}$$

This implies the following for all $k \geq 0$:

$$m_{k+1} - \nabla f(x_k) = (1-\alpha)^{k+1}(m_0 - \nabla f(x_0)) + \sum_{i=0}^{k}\alpha(1-\alpha)^{k-i}(g(\overline{x}_i, \xi_i) - \nabla f(\overline{x}_i))$$

$$+ \sum_{i=0}^{k-1}(1-\alpha)^{k-i-1}(\alpha\nabla f(\overline{x}_{i+1}) + (1-\alpha)\nabla f(x_i) - \nabla f(x_{i+1})).$$

Using this, we can upper-bound $\mathbb{E}\left[\|m_{k+1} - \nabla f(x_k)\|_*\right]$ for $k \geq 0$ as follows:

$$
\begin{aligned}
\mathbb{E}\left[\|m_{k+1} - \nabla f(x_k)\|_*\right] &\overset{(a)}{\leq} (1-\alpha)^{k+1}\mathbb{E}\left[\|m_0 - \nabla f(x_0)\|_*\right] \\
&\quad + \sum_{i=0}^{k-1}(1-\alpha)^{k-i-1}\|\alpha\nabla f(\overline{x}_{i+1}) + (1-\alpha)\nabla f(x_i) - \nabla f(x_{i+1})\|_* \\
&\quad + \mathbb{E}\left[\|\sum_{i=0}^{k}\alpha(1-\alpha)^{k-i}(g(\overline{x}_i, \xi_i) - \nabla f(\overline{x}_i))\|_*\right] \\
&\overset{(b)}{\leq} (1-\alpha)^{k+1}\rho\mathbb{E}\left[\|m_0 - \nabla f(x_0)\|_2\right] \\
&\quad + \sum_{i=0}^{k-1}(1-\alpha)^{k-i-1}\|\alpha\nabla f(\overline{x}_{i+1}) + (1-\alpha)\nabla f(x_i) - \nabla f(x_{i+1})\|_* \\
&\quad + \rho\mathbb{E}\left[\|\sum_{i=0}^{k}\alpha(1-\alpha)^{k-i}(g(\overline{x}_i, \xi_i) - \nabla f(\overline{x}_i))\|_2\right] \\
&\overset{(c)}{\leq} (1-\alpha)^{k+1}\rho\sqrt{\mathbb{E}\left[\|m_0 - \nabla f(x_0)\|_2^2\right]} \\
&\quad + \sum_{i=0}^{k-1}(1-\alpha)^{k-i-1}\|\alpha\nabla f(\overline{x}_{i+1}) + (1-\alpha)\nabla f(x_i) - \nabla f(x_{i+1})\|_* \\
&\quad + \rho\sqrt{\mathbb{E}\left[\|\sum_{i=0}^{k}\alpha(1-\alpha)^{k-i}(g(\overline{x}_i, \xi_i) - \nabla f(\overline{x}_i))\|_2^2\right]} \\
&\overset{(d)}{\leq} (1-\alpha)^{k+1}\rho\sigma + \alpha\rho\sigma\sqrt{\sum_{i=0}^{k}(1-\alpha)^{2(k-i)}} \\
&\quad + \sum_{i=0}^{k-1}(1-\alpha)^{k-i-1}\|\alpha\nabla f(\overline{x}_{i+1}) + (1-\alpha)\nabla f(x_i) - \nabla f(x_{i+1})\|_* \\
&\overset{(e)}{\leq} (1-\alpha)^{k+1}\rho\sigma + \sqrt{\alpha}\rho\sigma \\
&\quad + \sum_{i=0}^{k-1}(1-\alpha)^{k-i-1}\|\alpha\nabla^2 f(x_{i+1})(\alpha\overline{x}_{i+1} + (1-\alpha)x_i - x_{i+1})\|_* \\
&\quad + \sum_{i=0}^{k-1}\frac{H(1-\alpha)^{k-i-1}}{2}\left(\alpha\|\overline{x}_{i+1} - x_{i+1}\|^2 + (1-\alpha)\|x_i - x_{i+1}\|^2\right) \\
&\overset{(f)}{=} (1-\alpha)^{k+1}\rho\sigma + \sqrt{\alpha}\rho\sigma + \sum_{i=0}^{k-1}\frac{H(1-\alpha)^{k-i}}{2\alpha}\|x_i - x_{i+1}\|^2 \\
&\overset{(g)}{\leq} (1-\alpha)^{k+1}\rho\sigma + \sqrt{\alpha}\rho\sigma + \sum_{i=0}^{k-1}\frac{H\eta^2(1-\alpha)^{k-i}}{2\alpha} \\
&\leq (1-\alpha)^{k+1}\rho\sigma + \sqrt{\alpha}\rho\sigma + \frac{H\eta^2}{2\alpha^2},
\end{aligned}
$$

where (a) uses the triangle inequality; (b) uses Assumption (A3); (c) uses Jensen's inequality; (d) uses Assumption (A1) and the fact that samples $\xi_i \sim \mathcal{D}$ are i.i.d.; (e) uses the triangle inequality and Assumption (A5); (f) uses eq. (30); (g) uses eq. (29). $\qquad\square$

### D.6  PROOF OF THEOREM 6

In this proof, we are going to use Lemma 6, which are valid for the iterations of Algorithm 3. We also obtain the following Lemma 9. The proof is almost identical to the proof of Lemma 8 in Appendix D.5.1, with the only difference being that we use the inequality $\|x_{k+1} - x_k\| \leq 2\eta$, which holds due to Lemma 6.

**Lemma 9.** *Let Assumptions (A1), (A3) and (A5) hold, and let $x_0 \in \mathrm{dom}\, R$ and $m_0 = g(x_0, \xi_0)$. Then the iterations of Algorithm 3 satisfy the following inequality for $k \geq 0$:*

$$\mathbb{E}\left[\|m_{k+1} - \nabla f(x_k)\|_*\right] \leq (1-\alpha)^{k+1}\rho\sigma + \sqrt{\alpha}\rho\sigma + \frac{2H\eta^2}{\alpha^2}. \tag{65}$$

Now, we are ready to prove Theorem 6. Similarly to the proof of Theorem 4 in Appendix D.4, we can upper-bound $F(x_{k+1}) - F(x^*)$ as follows:

$$F(x_{k+1}) - F(x^*) \leq (1-\beta)(F(x_k) - F(x^*)) + 4L\eta^2 + 2\eta\|m_{k+1} - \nabla f(x_k)\|_*,$$

Using Lemma 8, we obtain the following inequality:

$$\mathbb{E}\left[F(x_{k+1}) - F(x^*)\right] \leq (1-\beta)\mathbb{E}\left[F(x_k) - F(x^*)\right] + 2\eta\rho\sigma(1-\alpha)^k + 2\eta\rho\sigma\sqrt{\alpha}$$
$$+ 4L\eta^2 + \frac{4H\eta^3}{\alpha^2},$$

which implies the following inequality:

$$\mathbb{E}\left[F(x_K) - F(x^*)\right] \leq (1-\beta)^K(F(x_0) - F(x^*)) + 2\eta\rho\sigma\left(\frac{1}{\alpha} + \frac{\sqrt{\alpha}}{\beta}\right) + \frac{4L\eta^2}{\beta} + \frac{4H\eta^3}{\alpha^2\beta}.$$

$\square$

### D.7  PROOF OF THEOREM 7

We start with the following inequality obtained in the proof of Theorem 1 in Appendix D.1:

$$F(x_{k+1}) \leq F(x_k) - \eta\|\nabla f(x_{k+1}) + \hat{\nabla}R_{k+1}\|_* + \tfrac{3}{2}L\eta^2. \tag{66}$$

Using this inequality, we can upper-bound $F(x_{k+1})$ as follows:

$$\begin{aligned}
F(x_{k+1}) &\stackrel{(a)}{=} f(x_{k+1}) + R(x_{k+1}) \\
&\stackrel{(b)}{\leq} f(x^*) + R(x^*) + \langle \nabla f(x_{k+1}) + \hat{\nabla}R_{k+1}, x_{k+1} - x^* \rangle \\
&\stackrel{(c)}{=} F(x^*) + \langle \nabla f(x_{k+1}) + \hat{\nabla}R_{k+1}, x_{k+1} - x^* \rangle \\
&\stackrel{(d)}{\leq} F(x^*) + \|x_{k+1} - x^*\|\|\nabla f(x_{k+1}) + \hat{\nabla}R_{k+1}\|_* \\
&\stackrel{(e)}{\leq} F(x^*) + D\|\nabla f(x_{k+1}) + \hat{\nabla}R_{k+1}\|_* \\
&\stackrel{(f)}{\leq} F(x^*) + (D/\eta)(F(x_k) - F(x_{k+1})) + \tfrac{3}{2}LD\eta,
\end{aligned}$$

where (a) and (c) use the definition of $F(x)$ in problem (1); (b) uses the convexity of function $R(x)$ and Assumption (A4); (d) uses the definition of the dual norm; (e) uses Assumption (A6) and the fact that $x_{k+1}, x^* \in \mathrm{dom}\, R$; (f) uses eq. (66). After rearranging, we obtain

$$F(x_{k+1}) - F(x^*) \leq \frac{D}{\eta + D}\left(F(x_k) - F(x^*) + \frac{3L\eta^2}{2}\right),$$

which implies the following inequality:

$$F(x_K) - F(x^*) \leq \left(\frac{D}{\eta + D}\right)^K (F(x_0) - F(x^*)) + \frac{3LD\eta}{2}.$$

$\square$

## D.8 PROOF OF THEOREM 8

Similarly to the proof of Theorem 7 in Appendix D.7, we can obtain the following inequality:

$$F(x_{k+1}) \leq F(x^*) + D\|\nabla f(x_{k+1}) + \hat{\nabla} R_{k+1}\|_*. \tag{67}$$

Combining this inequality with Lemma 4 gives the following:

$$F(x_{k+1}) - F(x^*) \leq \frac{D}{\eta + D}\left(F(x_k) - F(x^*) + 2\eta\|\nabla f(x_{k+1}) - m_{k+1}\|_* + \frac{3L\eta^2}{2}\right)$$

$$\overset{(a)}{\leq} \frac{D}{\eta + D}\left(F(x_k) - F(x^*) + 2\eta\|\nabla f(x_k) - m_{k+1}\|_* + \frac{7L\eta^2}{2}\right),$$

where (a) uses the triangle inequality, Assumption (A2), and eq. (15). After taking the expectation and using Lemma 5, we obtain the following:

$$\mathbb{E}\left[F(x_{k+1}) - F(x^*)\right] \leq \left(\frac{D}{\eta + D}\right)\mathbb{E}\left[F(x_k) - F(x^*)\right] + \frac{2L\eta^2}{\alpha} + \frac{3L\eta^2}{2}$$

$$+ 2(1-\alpha)^{k+1}\eta\rho\sigma + 2\sqrt{\alpha}\eta\rho\sigma,$$

which implies the following inequality:

$$\mathbb{E}\left[F(x_K) - F(x^*)\right] \leq \left(\frac{D}{\eta + D}\right)^K (F(x_0) - F(x^*)) + 2\sqrt{\alpha}D\rho\sigma + \frac{2\eta\rho\sigma}{\alpha} + \frac{3LD\eta}{2} + \frac{2LD\eta}{\alpha}.$$

$$\square$$

