# OpenReview forum: "Understanding Gradient Orthogonalization for Deep Learning via Non-Euclidean Trust-Region Optimization"
_ICLR.cc/2026/Conference — Submitted to ICLR 2026_

### Official Review · Reviewer_8eaq · 2025-10-29

**Soundness:** 2
**Presentation:** 3
**Contribution:** 2
**Rating:** 2
**Confidence:** 3

**Summary:**

This paper interprets Muon as an exact non-Euclidean trust-region (TR) method and develops stochastic TR algorithms (with momentum, weight decay, and an extrapolation variant) for recovering Muon-style updates. It proves non-convex convergence and faster rates under star-convexity and second-order smoothness. In addition, this paper offers some explanations for the comparison between Muon and OSGDM.

**Strengths:**

1. This paper provides a unifying perspective of recent optimizers by casting orthogonalized/spectral updates as exact trust-region steps in a chosen norm.
2. The theoretical analyses explicitly cover momentum, decoupled weight decay (as center-shifting in TR), and an extrapolation option, with the inclusion of nonconvex (deterministic and stochastic), star-convex, and second-order smoothness settings.
3. The paper offers some insights on why Muon can outperform OSGDM, which could probably be highlighted in the main text.

**Weaknesses:**

1. The major concern is the dimension dependence hidden in Assumption A3: treating the norm-equivalence constant $\rho$ as a universal constant seems unrealistic for non-Euclidean norms, as $\rho$ could be as large as $\Theta(\sqrt{d})$. In high-dimensional optimization theory,  $d \in \mathcal{O}(poly(T))$, which would invalidate the claimed $O(\varepsilon^{-4})$ and challenge the theoretical rigor and significance of the analysis.
2. The “norm constraint” perspective (in contrast to the steepest descent in non-Euclidean norm view by [Bernstein& Newhouse, 2024]) has appeared in earlier work on spectral/normalized steepest descent (and Muon) from an implicit-bias angle [4], but is not mentioned in the paper. It would be good for the authors to clarify why the trust-region framework differs from the ball-constraint framework.
3. The paper claims to be the first convergence analysis for Muon, but yet lacks sufficient comparison against previous works on the convergence of Muon/spectral-constraint updates—Li & Hong [1], Shen et al. [2], Chen et al. [3].
4. Based on the above, the trust-region–based analysis, while comprehensive in well-studied settings such as star-convex and second-order smooth objectives, offers relatively limited novelty and theoretical contribution in the context of analyzing the Muon optimizer.
5. Minor: Typos at lines 94 and 808–809.


References:

[1] Li & Hong. “A Note on the Convergence of Muon.” arXiv:2502.02900, 2025.

[2]Shen et al. “On the Convergence Analysis of Muon.” arXiv:2505.23737, 2025.

[3] Chen et al. “Muon Optimizes Under Spectral Norm Constraints.” arXiv:2506.15054, 2025.

[4] Fan et al. “Implicit Bias of Spectral Descent and Muon on Multiclass Separable Data” arXiv:2502.04664, 2025.

**Questions:**

See weaknesses.

---

### Official Review · Reviewer_RgWt · 2025-10-30

**Soundness:** 3
**Presentation:** 2
**Contribution:** 3
**Rating:** 6
**Confidence:** 3

**Summary:**

This work investigates a first-order trust-region gradient method in a non-Euclidean setting under various assumptions. In particular, the authors demonstrate that gradient orthogonalization, as used in the Muon optimizer, can be viewed as a specific instance of their framework. They further establish convergence guarantees for multiple scenarios: the non-stochastic non-convex case without momentum, both stochastic and non-stochastic non-convex cases with momentum, the stochastic star-convex case with momentum and weight decay, and the stochastic non-convex and star-convex cases incorporating momentum, weight decay, and extrapolation.

**Strengths:**

- The paper gives a potential explanation of why weight decay is not necessary in training small-scale language models as opposed to larger ones.
- The analyses encompass both stochastic and non-convex formulations, including non-convex, star-convex and second-order smooth functions, with convergence guarantees established for each setting.
- The paper gives potential explanations on why Muon outperforms Orthogonal-SGDM in practice.

**Weaknesses:**

- The convergence rate of the weight decay variants is only provided under additional assumptions (star-convexity or second-order smoothness).
- The paper would benefit from a more detailed discussion of choices of regularization (aside from weight clipping), its interplay with weight decay as an implicit regularizer, and how these choices affect the optimization dynamics.
- I think there are a few other works on analyzing the convergence of Muon that should also be cited.
- Many arguments are introduced with phrases such as "It is easy to verify that", or "one can show that", which require the reader to fill in significant details to verify them (e.g., lines 342-345, 348-350, 966-968).
- Some key theoretical explanations for Muon’s performance are presented in the Appendix rather than in the main text (e.g.  Muon vs Orthogonal-SGDM in Appendix B,  Algorithm 1 with weight clipping vs Algorithm 2 with weight decay in Appendix C).
- I understand that this is a theory paper, but it would be nice to include some experimental results, for example on the new variant of Muon with extrapolation and weight decay, or potentially on some other algorithms within the Non-Euclidean Trust-Region framework, to verify the improved convergence rates experimentally, especially since the title explicitly mentions "deep learning".

**Questions:**

- Would the algorithms benefit from using a trust-region radius that adapts across iterations? Have the authors considered that?
- Is the inclusion of the weight decay term and thus the boundedness of the iterates the reason why the complexity of Corollary 4 matches the complexity of Corollary 8?

---

### Official Review · Reviewer_nGpf · 2025-10-31

**Soundness:** 3
**Presentation:** 3
**Contribution:** 2
**Rating:** 4
**Confidence:** 4

**Summary:**

This paper studies the theoretical analysis of a class of optimization algorithms, including Muon, SignSGD, and normalized SGD. The authors established convergence analysis under smooth nonconvex and star-convex settings, with standard assumptions on stochastic gradient noise.

**Strengths:**

1. The paper is well-written, with all the concepts and notations explained clearly. I enjoy reading it.
2. The convergence results of the framework are clean and easy to follow. The star-convex convergence of algorithms with weight decay looks interesting, may help better understand the role of weight decay in training.

**Weaknesses:**

1. The trust-region analysis framework, which the authors claimed as one of the major contributions, seems to coincide with the  LMO framework presented in [1]. If that is considered as a concurrent work, I think the authors should at least cite the paper and discuss the difference with it.
2. The main theoretical results in the nonconvex parts doesn't seem to have enough novelty for me. Baiscally, the analysis in both first-order smooth and second-order smooth settings follows generally the same mainline as the Normalized SGD (with momentum) [2] and SignSGD (with momentum) [3] convergence analysis, which are well-established results. The variance-reduction effect of momentum for this kind of "normalized" optimizers  has also been discussed clearly in these existing works. This paper presents a more general framework to unite these results, and further includes a regularization term, which is still not novel enough technically.
3. Under the analysis framework, the convergence results rely on $ \rho $, which can heavily depend on the dimension of the problem. In a practical setting, $ \rho $ can be extremely large. Thus, this results at least fail to explain why Muon can perform better than SGD with momentum, as also mentioned by the authors.

**References**

[1] Pethick, Thomas, et al. "Training deep learning models with norm-constrained lmos." ICML 2025.

[2] Cutkosky, Ashok, and Harsh Mehta. "Momentum improves normalized sgd." ICML, 2020.

[3] Sun, Tao, et al. "Momentum ensures convergence of signsgd under weaker assumptions." ICML, 2023.

**Questions:**

1. Could you better explain why the convergence results of star-convex functions (Section 4.2) can explain the reason for that Muon needs weight decay more for larger models, as claimed in the contribution part?
2. If we assume the variance is bounded under $ ||\cdot||_{\star} $ instead of the Euclidean one (or maybe similar to the anisotropic variance assumption in [1]), could the convergence analysis of the framework get rid of the dependence of $ \rho $?

**References**

[1] An, Kang, et al. "Asgo: Adaptive structured gradient optimization." arXiv preprint arXiv:2503.20762 (2025).

---

### Official Review · Reviewer_hr2W · 2025-10-31

**Soundness:** 3
**Presentation:** 3
**Contribution:** 3
**Rating:** 6
**Confidence:** 3

**Summary:**

This paper presents matrix gradient orthogonalization from Muon optimizers as a non-Euclidean trust-region optimization technique, which uses matrix spectral norm as its definition. The new framework establishes the proof of convergence for Muon while explaining why weight decay techniques work for training big language models in practice.

**Strengths:**

1. Theoretical analysis shows why Muon achieves superior results in practice through its pre-orthogonalization momentum application compared to Orthogonal-SGDM, which applies momentum after orthogonalization.
2. Under the additional assumption of second-order smoothness, the paper develops an algorithm (Algorithm 3) that achieves an improved $1/\epsilon^{3.5}$ complexity for non-convex functions, which is better than the standard $1/\epsilon^4$ rate.

**Weaknesses:**

1. The theory did not address both step-by-step cost calculations and stability issues that occur when using rectangular or low-rank matrices during large-scale training operations.
2. The paper lacks empirical evidence to support its claims because it presents only theoretical content without conducting new experimental research. The untested hypothesis of Algorithm 3 lacks experimental verification.
3. The research achieves better convergence rates through assumptions that might not apply to actual deep learning systems. 1) The assumption about the geometry of the loss landscape is not universally accepted for complex neural networks.   2) The assumption of a Lipschitz continuous Hessian is an even stronger condition that is difficult to verify or justify for modern, large-scale architectures.

**Questions:**

See weakness.

---

### Meta-Review · Area_Chair_c2fu · 2025-12-24

**Summary:**

Simply put, the authors did not make an effort to respond to the reviewers’ questions or clarify the raised concerns. Therefore, this paper is rejected.

Nevertheless, the new version of this paper might need to address the following points:

- The authors should either elaborate on how the convergence results for star-convex functions (Section 4.2) explain the reason for that Muon might need weight decay more for larger language models, as claimed in the contribution part (e.g., the one in the abstract), or revise to tone down and be precise.

- Three out of four reviewers raised concerns or questions regarding the positioning of this work within the existing literature on Muon. Thus, a more detailed and careful comparison is necessary. A more comprehensive literature review related to Muon should be included. This is particularly important because the convergence rates for the weight-decay variants are established only under additional assumptions (e.g., star-convexity or second-order smoothness), and that the norm-constraint perspective (as opposed to the steepest-descent-in-non-Euclidean-norm view of Bernstein & Newhouse, 2024) has appeared in earlier work on spectral or normalized steepest descent, but is not sufficiently discussed in the paper as raised by a couple of reviewers.

- The paper would benefit from a more detailed discussion of regularization choices beyond weight clipping, their interaction with weight decay as an implicit regularizer, and how these design choices influence the optimization dynamics.

**Reviewer Concerns:**

N/A, since the authors did not make an effort to clarify the raised concerns.

**Reviewer Scores:**

N/A, since the authors did not make an effort to clarify the raised concerns.

---

### Decision · Program_Chairs · 2026-01-26

Reject